# Graph4MM: Weaving Multimodal Learning with Structural Information

**Xuying Ning** [* 1]  **Dongqi Fu** [* 2]  **Tianxin Wei** [1]  **Wujiang Xu** [3]  **Jingrui He** [1]

## Abstract

Real-world multimodal data usually exhibit complex structural relationships beyond traditional one-to-one mappings like image-caption pairs. Entities across modalities interact in intricate ways, with images and text forming diverse interconnections through contextual dependencies and co-references. Graphs provide powerful structural information for modeling intra-modal and inter-modal relationships. However, previous works fail to distinguish multi-hop neighbors and treat the graph as a standalone modality, which fragments the overall understanding. This limitation presents two key challenges in multimodal learning: (1) integrating structural information from multi-hop neighbors into foundational models, and (2) fusing modality-specific information in a principled manner. To address these challenges, we revisit the role of graphs in multimodal learning within the era of foundation models and propose Graph4MM, a graph-based multimodal learning framework. To be specific, we introduce Hop-Diffused Attention, which integrates multi-hop structural information into self-attention through causal masking and hop diffusion. Furthermore, we design MM-QFormer, a multi-mapping querying transformer for cross-modal fusion. Through theoretical and empirical analysis, we show that *leveraging structures to integrate both intra- and inter-modal interactions improves multimodal understanding beyond treating them as a standalone modality*. Experiments on both generative and discriminative tasks show that Graph4MM outperforms larger VLMs, LLMs, and multimodal graph baselines, achieving a 6.93% average improvement.

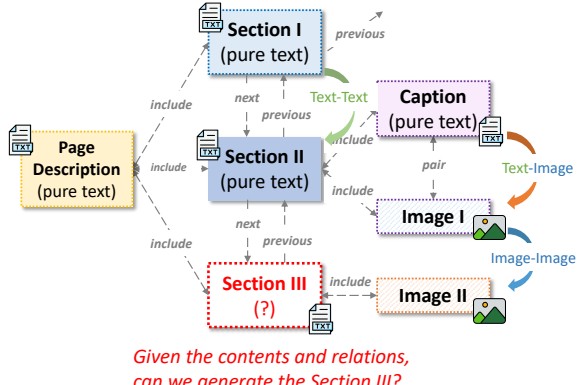

*Figure 1.* Multimodal relationships in a document, where sections, images, captions, and page descriptions form a structured graph. The task aims to generate Section III using the neighboring context.

## 1. Introduction

Data modalities, in general, refer to different types of the data, like text, image, and audio (Ngiam et al., 2011). Jointly considering cross-modality data has great research and application potential, and it has been widely applied in image generation, question answering, and many more (Gao et al., 2020; Summaira et al., 2021; Stahlschmidt et al., 2022; Jabeen et al., 2023). Recently, a pioneering study (Yoon et al., 2023) discovered that the relationship of data modalities is complex and far beyond the one-to-one modeling, like the image-caption pair, in real-world scenarios. For example, in an academic paper, the image and text data have different semantic and non-linear correlations, i.e., the image and its caption have a direct pairing relation to each other, but the relation between the image and its following section contents and the page summarization is intricate, as shown in Figure 1. However, most Vision-Language Models (VLMs) (Alayrac et al., 2022a; Li et al., 2023c; Koh et al., 2023; Zhang et al., 2024a) are still limited to modeling one-to-one relationships between images and text, making them inadequate for capturing complex multimodal interactions.

Graphs, as a relational data structure, have the inherent advantage of modeling complex modality relationships together. To the best of our knowledge, MMGL (Yoon et al., 2023) is the state-of-the-art work that models modalities into graphs and obtains promising performance in the generation task, as compared to single-modal pre-trained language models and vision-language models. In particular, it

---

*Equal contribution [1]University of Illinois Urbana-Champaign [2]Meta AI [3]Rutgers University. Correspondence to: Jingrui He <jingrui@illinois.edu>.

*Proceedings of the 42nd International Conference on Machine Learning*, Vancouver, Canada. PMLR 267, 2025. Copyright 2025 by the author(s).

models each modality data item (e.g., a sentence piece, an image) as a node and establishes their edges based on the concrete application scenario. For example, when addressing section summarization, edges are established based on section-subsection hierarchy and co-occurrence of images and text within the same section.

Establishing the modality graph can help select and fuse the important knowledge from the raw multimodal data and further contribute to solving the context length limitation for language models. Nevertheless, the modality fusion in the pioneering MMGL (Yoon et al., 2023) is simple and does not fully exploit the complex relationship. To be more specific, although cross-modal data is modeled into multimodal graphs, MMGL simply concatenates the neighbors's multi-modal data together, where the internal adjacency is largely ignored, e.g., different distant nodes are treated equally. Moreover, MMGL regards the established graph as a standalone modality with images and text. However, this approach for injecting graph topology information does not yield the expected performance improvements.

Motivated by the above observation, we further explore *the role of graphs in multimodal learning in the era of foundation models* from empirical and theoretical analysis in Section 4.3. Briefly, unlike conventional modalities such as text and images with densely pretrained representation, treating graph structures as an independent modality and projecting graph embeddings into the same space as language and vision models often result in suboptimal performance. This is primarily due to the volume of training data and the well-aligned feature spaces of pre-trained language and vision foundation models, which can hinder effective fusion and limit the model's ability to do downstream tasks.

Therefore, we propose **Graph4MM**, a structured multimodal learning framework that simultaneously captures intra-modal multi-hop structural connectivity and fuses inter-modal representations in a principled manner. Our contributions are summarized as follows:

● **A Novel Structure-Guided Paradigm for Multimodal Learning.** To address the limitations of existing multimodal learning methods in capturing complex modality interactions, we propose Graph4MM, a multimodal learning framework that integrates structural information from multihop neighbors into foundation models and fuses modality-specific representations in a principled manner.

● **Designs of Hop-Diffused MM-QFormer.** We introduce Hop-Diffused Attention, which incorporates multi-hop connectivity into self-attention using causal masking and hop diffusion. Theoretical analysis shows that it avoids oversmoothing and does not rely on stacking multiple GNN layers for multi-hop aggregation. Additionally, we design MM-QFormer, a querying transformer, to facilitate cross-

modal fusion.

● **Revisiting the Role of Graphs in Multimodal Learning.** We conduct both theoretical and empirical analysis on the role of graphs in multimodal learning within the era of foundation models. Our findings suggest that leveraging topological structures to guide intra- and inter-modal interactions is more effective than treating graphs as an independent modality.

● **State-of-the-Art Performance Across Generation and Discrimination.** We evaluate Graph4MM on generative (e.g., academic paper section summarization) and discriminative (e.g., zero-shot fashion classification) tasks. Extensive experiments show that it consistently outperforms pretrained VLMs, LLMs, and multimodal graph learning methods, achieving an average improvement of 6.93% across textual and visual metrics.

## 2. Preliminary

In this section, to pave the way for introducing our multimodal framework Graph4MM, we first introduce the multimodal graph modeling and the corresponding advantages. Then, we introduce two formal tasks the proposed Graph4MM aims to solve.

### 2.1. Multimodal Graph Modeling

Existing works (Yoon et al., 2023; Jin et al., 2024; Zhu et al., 2024b) offer varying definitions of multimodal graphs. To unify these perspectives, we define a multimodal graph as an unweighted, undirected structure: $\mathcal{G} = (\mathcal{V}, \mathcal{E}, \mathcal{T}, \mathcal{P})$.

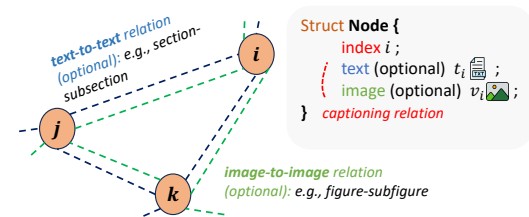

*Figure 2.* Multimodal Graph Modeling

**Node**. In the multimodal graph $\mathcal{G} = (\mathcal{V}, \mathcal{E}, \mathcal{T}, \mathcal{P})$, we model a node with a unique index and optional textual and visual attributes. Mathematically, each node $v_i \in \mathcal{V}$ (e.g., a section in a webpage) is represented by a unique node index $i$ and may optionally include textual attribute $t_{v_i} \in \mathcal{T}$ (e.g., its textual content) and visual attribute $p_{v_i} \in \mathcal{P}$ (e.g., an associated image).

**Edge**. In the multimodal graph $\mathcal{G} = (\mathcal{V}, \mathcal{E}, \mathcal{T}, \mathcal{P})$, we consider three kinds of edges, i.e., *text-to-text*, *image-to-image*, and *text-to-image*. Based on our node modeling, the text-to-image relationship, like captioning, is basic and already encoded in each node. Further, the text-to-text edge exists

between nodes $i$ and $j$ if and only if both nodes contain non-null text features and share a meaningful predefined real-world relationship (e.g., section-subsection in a paper, descriptions of frequently co-purchased items). The same rule also applies to image-to-image edges between two nodes.

Through this multimodal graph representation, for a node $i$, along text-to-text (or image-to-image) edges, we can easily induce its *textual subgraph* $\mathcal{G}_t = (\mathcal{V}_t, \mathcal{E}_t, \mathcal{T}_t, \mathcal{P}_t)$ as $\mathcal{V}_t = \{u \in \mathcal{V} \mid \text{dist}(u, v_i) \leq \tau\}$, where $\mathcal{V}_t$ includes nodes with textual attributes reachable within $\tau$-hops, and $\mathcal{E}_t$ captures text-text relationships. Similarly, the *visual subgraph* $\mathcal{G}_p = (\mathcal{V}_p, \mathcal{E}_p, \mathcal{T}_p, \mathcal{P}_p)$ models image-to-image interactions. These subgraphs, combined with text-image associations at the node level, act as adjacency heuristics that guide causal masking attention, facilitating both intra-modality coherence and inter-modality fusion.

## 2.2. Task Definitions

By modeling multiple modalities into graphs, we then mathematically introduce the tasks Graph4MM aims to solve.

**Generation Task**. Given a node $v_i \in \mathcal{V}$, an instruction prompt $\mathcal{I}$, and its associated multimodal subgraphs $\mathcal{G}_t$ and $\mathcal{G}_p$, our objective is to generate a response $\hat{\mathbf{y}}_{v_i} = \{\hat{y}_1, \hat{y}_2, \ldots, \hat{y}_L\}$ by maximizing:

$$\hat{\mathbf{y}}_{v_i} = \arg\max_{\mathbf{y}_{v_i}} \prod_{t=1}^{L} P(y_t \mid \hat{y}_1, \hat{y}_2, \ldots, \hat{y}_{t-1}, \mathcal{G}_t, \mathcal{G}_p, \mathcal{I}), \quad (1)$$

where $\mathcal{G}_t$ and $\mathcal{G}_p$ represent the textual and visual subgraphs. The generated response leverages multimodal interactions to address the task defined by $\mathcal{I}$. For example, in document understanding, given the instruction "Summarize this section", Graph4MM integrates broader page-level multimodal context to generate a coherent and context-aware summary.

**Discriminative Task**. Here, we target the more challenging zero-shot classification task, where an LLM is called to generate a language response conditioned on the textual subgraph $\mathcal{G}_t$, the visual subgraph $\mathcal{G}_p$, and a classification prompt $\mathcal{I}$. For a given node $v_i \in \mathcal{V}$, LLM generates a response $\hat{\mathbf{y}}_{v_i}$ for the classification answer and assigns the node to the most relevant class by solving:

$$c(v_i) = \arg\max_{c_j} \phi(\hat{\mathbf{y}}_{v_i}, \mathbf{d}_{c_j}), \quad (2)$$

where $\hat{\mathbf{y}}_{v_i} = \text{PLM}(\mathcal{G}_t, \mathcal{G}_p, \mathcal{I})$ is the generated response from the pretrained language model, and $\phi(\cdot)$ denotes a similarity function. Here, $\mathbf{d}_{c_j}$ represents the embedding of the description of class $j$. Graph4MM utilizes both node and neighbor attributes for enhanced zero-shot classification, which is generally unavailable in traditional graph learning.

## 3. Proposed Graph4MM Framework

In this section, we systematically present our Graph4MM framework, which synergizes the heterogeneous inputs from vision and language while simultaneously capturing their complex structural interactions. The overview of our framework is presented in Figure 3. To be specific, in Subsection 3.1, we first introduce an elementary backbone for fusing multimodal information, which serves as a vanilla architecture to inspire the more effective Graph4MM. Then, in Subsection 3.2, we dive into each modality and utilize multi-hop neighbor connectivity in the graph to guide the intra-modality information fusion via the proposed Hop-Diffused Attention. Finally, in Subsection 3.3, we introduce multimodal query transformer, MM-QFormer, which captures inter-modal interaction and fuses modal-specific information for downstream foundation models to deal with generation and discrimination tasks.

## 3.1. The Vanilla Architecture for Fusing Multimodal Information

The emergent reasoning capabilities (Wei et al., 2022a) of large language models are essential for tackling zero-shot open-ended QA tasks (Guo et al., 2023). To effectively preserve these abilities of pretrained language models while efficiently integrating relevant visual information, our multimodal fusion strategy focuses on projecting visual inputs seamlessly into the LLM's semantic space, leveraging off-the-shelf frozen pretrained language and vision models.

**Textual Context Integration.** We define the textual context $c_T(\mathcal{G}_t)$ derived from the subgraph $\mathcal{G}_t = (\mathcal{V}_t, \mathcal{E}_t, \mathcal{T}_t, \mathcal{P}_t)$, where $\mathcal{V}_t$ consists of the target node $v_i$ and its $\tau$-hop neighbors $\mathcal{N}(v_i) = \{u \in \mathcal{V} \mid \text{dist}(u, v_i) \leq \tau\}$. The textual input sequence is constructed as:

$$c_T(\mathcal{G}_t) = \left[ h_T(I), \{h_T(t_v) \mid v \in \mathcal{V}_t\} \right] \in \mathbb{R}^{l_T \times d}, \quad (3)$$

where $I$ represents instruction prompts, $h_T(\cdot)$ maps text inputs to their corresponding token embeddings, $l_T$ is the sequence length of the complete textual input, $d$ is the dimensionality of the token embeddings, and $[\cdot]$ represents the concatenation operation.

**Visual Context Integration.** The visual context $c_P(\mathcal{G}_p)$ is derived from the visual attributes of nodes in $\mathcal{G}_p = (\mathcal{V}_p, \mathcal{E}_p, \mathcal{T}_p, \mathcal{P}_p)$. Using a frozen visual encoder $g_{\text{img}}(\cdot)$, we compute the visual embeddings as:

$$c_P(\mathcal{G}_p) = \left[ \{g_{\text{img}}(p_v) \mid v \in \mathcal{V}_p\} \right] \in \mathbb{R}^{|\mathcal{V}_p| \times d_p}, \quad (4)$$

where $p_v \in \mathcal{P}_p$ denotes the visual attribute of node $v$, and $d_p$ is the dimensionality of the visual embeddings.

**Structured Multi-modal Input for PLM.** Both textual and visual contexts are processed separately and then con-

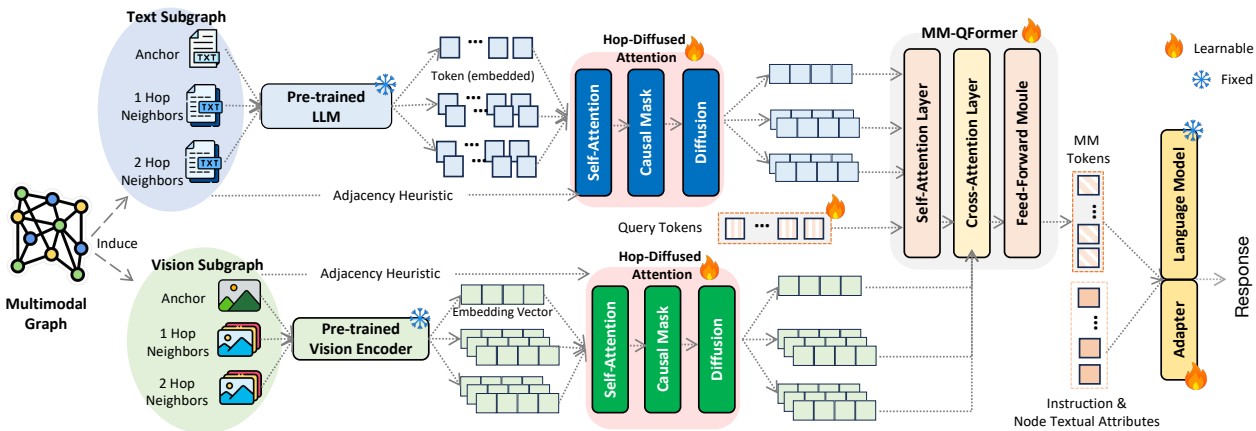

*Figure 3.* Overview of the Graph4MM framework. Hop-Diffused Attention applies adjacency heuristics via causal masking and hop diffusion, while MM-QFormer captures fine-grained text-image interactions before passing inputs to the LLM.

catenated to form the input to the pretrained language model:

$$h(c_T, c_P) = \left[ c_T(\mathcal{G}_t), \mathbf{f}(c_P(\mathcal{G}_p)) \right] \in \mathbb{R}^{(l_T + l_P) \times d},$$

where $c_T(\mathcal{G}_t) \in \mathbb{R}^{l_T \times d}$ represents the textual context, and $\mathbf{f}(c_P(\mathcal{G}_p)) \in \mathbb{R}^{l_P \times d}$ maps the visual context $c_P(\mathcal{G}_p) \in \mathbb{R}^{|\mathcal{V}_p| \times d_p}$ into the PLM's semantic space using a transformation function $\mathbf{f} : \mathbb{R}^{|\mathcal{V}_p| \times d_p} \rightarrow \mathbb{R}^{l_P \times d}$, which will be comprehensively introduced in the Section 3.3. This mapping ensures the alignment of visual features with the token embeddings of the language model, and $l_P$ is the number of multi-modal (MM) tokens corresponding to each node. Then, the PLM takes $h(c_T, c_P)$ as input tokens and generates target outputs based on the task requirements as mentioned in the instruction.

In our implementation, the MM tokens corresponding to each node are inserted directly after the textual attribute tokens of the same node. This arrangement ensures adjacency between textual and MM tokens, effectively highlighting the one-to-one correspondence for downstream pretrained language models to process multimodal inputs.

### 3.2. Hop-Diffused Attention.

Existing vision-language models (Li et al., 2023b; Yuan et al., 2021; Tsimpoukelli et al., 2021; Radford et al., 2021) primarily align single image-text pairs, overlooking the complex relationships among multiple images and text in tasks like ours. For example, LLMs often treat all visual inputs equally, ignoring the greater importance of a node's own image. To address this, we propose Hop-Diffused Attention which integrates graph structure form multi-hop connectivity into node embeddings through self-attention, causal masking, and a diffusion mechanism.

**Text and Vision Encoding.** We encode the text attributes of nodes in $\mathcal{V}_t$ using a frozen pretrained language model. Mean pooling over the sequence length dimension produces

textual embeddings: $\mathbf{H}_T \in \mathbb{R}^{|\mathcal{V}_t| \times d}$. To integrate visual information, visual embeddings are first concatenated over all nodes in $\mathcal{V}_p$, and then projected into the token embedding space of the language model via a learnable linear transformation $\mathbf{M}_{\text{proj}} \in \mathbb{R}^{d_p \times d}$: $\mathbf{H}_P = \bigoplus_{v \in \mathcal{V}_p} g_{\text{img}}(p_v) \cdot \mathbf{M}_{\text{proj}} \in \mathbb{R}^{|\mathcal{V}_p| \times d}$. The Hop-Diffused Attention module is completely symmetric for both visual embeddings $\mathbf{H}_P$ and textual embeddings $\mathbf{H}_T$; for brevity, we use $\mathbf{H}_P$ to illustrate below.

**Self-Attention.** Self-attention captures semantic relationships between node embeddings. For $\mathbf{H}_P \in \mathbb{R}^{|\mathcal{V}_p| \times d}$, where each row $\mathbf{h}_{v_i}$ represents the embedding of node $v_i$, and $\mathbf{q}_{v_i} = \mathbf{W}_q \mathbf{h}_{v_i}$, $\mathbf{k}_{v_j} = \mathbf{W}_k \mathbf{h}_{v_j}$, the initial attention matrix $\mathbf{A}' \in \mathbb{R}^{|\mathcal{V}_p| \times |\mathcal{V}_p|}$ is computed as:

$$\mathbf{A}'_{i,j} = \text{Softmax}_j \left( \frac{\mathbf{q}_{v_i}^\top \mathbf{k}_{v_j}}{\sqrt{d}} \right), \tag{5}$$

where $\mathbf{W}_q, \mathbf{W}_k \in \mathbb{R}^{d \times d}$ are learnable weight matrices projecting node embeddings into query and key spaces, and $\mathbf{A}'_{i,j}$ reflects the importance of node $v_j$ to node $v_i$.

**Causal Masking.** To encode the graph structure, we define a causal mask $\mathbf{M}_{i,j}$ that restricts attention to valid neighbors based on $\mathcal{E}_p$, the edge set of the subgraph $\mathcal{G}_p$. The mask is defined as:

$$\mathbf{M}_{i,j} = \begin{cases} 1, & \text{if } (v_i, v_j) \in \mathcal{E}_p, \\ 0, & \text{otherwise}. \end{cases} \tag{6}$$

This mask ensures that attention is computed only for nodes connected by edges in $\mathcal{E}_p$. The attention scores between nodes are then calculated as: $\mathbf{A}_{i,j} = \text{Softmax}(\mathbf{M}_{i,j} \cdot \mathbf{A}'_{i,j})$. By applying $\mathbf{M}_{i,j}$, nodes are restricted to attend only to their immediate neighbors, preserving the graph's local structure. This ensures that the attention mechanism aligns with the graph topology and the downstream model can distinguish between semantically similar images with different connectivity (e.g., directly connected or not connected).

**Diffusion Mechanism.** While causal masking ensures local connectivity, multi-hop structural information is captured via a diffusion mechanism that propagates attention across neighbors iteratively. Inspired by Wang et al. (2020), the diffusion matrix $\mathcal{A}$ is defined as:

$$\mathcal{A} = \sum_{i=0}^{\infty} \theta_i \mathbf{A}^i, \quad \theta_i = \alpha(1-\alpha)^i, \quad \alpha \in (0,1), \quad (7)$$

where $\mathbf{A}^i$ represents the $i$-power of the masked attention matrix $\mathbf{A}$, and $\theta_i$ are decay coefficients with $\sum_{i=0}^{\infty} \theta_i = 1$. The parameter $\alpha$ controls the influence of higher-hop neighbors, prioritizing closer connections through exponential decay. In practical implementation, we approximate the infinite summation by truncating at a finite number of steps $K$ referred to as the diffusion step.

The node embeddings are updated using the diffused attention matrix with residual connection:

$$\mathbf{H}_P \leftarrow \mathbf{H}_P + \mathcal{A}\mathbf{H}_P, \quad (8)$$

where $\mathbf{H}_P \in \mathbb{R}^{|\mathcal{V}_p| \times d}$ integrates multi-hop structural information into the node representations after the above update. This formulation enables nodes to aggregate information not only from their direct neighbors but also from distant nodes while attenuating the impact of farther nodes. The resulting hop-diffused embeddings $\mathbf{H}_P$ and $\mathbf{H}_T$ are then fed into the MM-QFormer, which will be discussed in the Section 3.3 ensuring effective multimodal fusion and structural information integration.

**Theoretical Justification of Hop-Diffused Attention.** We establish the validity of Hop-Diffused Attention by proving that the diffused attention matrix preserves key attention properties, ensuring that each row sums to one. This guarantees that the mechanism remains a valid attention operation while generalizing Personalized PageRank with an adaptive multi-hop weighting scheme. Detailed proof and theoretical analysis can be found in Appendix A and Appendix B.

We further analyze Hop-Diffused Attention in terms of *gradient smoothing* and demonstrate why it cannot be replaced by traditional graph models like GAT (Veličković et al., 2018). Specifically, we establish the following result:

**Proposition 3.1.** *Let $\mathbf{X}^{(1)}$ be the node representation matrix after the Hop-Diffused Attention module, and $\mathbf{X}^{(k)}$ be the representation at the $k$-th layer of GAT. When both methods aggregate information from $k$-hop neighbors, Hop-Diffused Attention retains higher Dirichlet energy, preserving more feature variance and mitigating over-smoothing. Formally, for large $k$,*

$$\mathcal{E}_{\text{Hop-Diffused}}(\mathbf{X}^{(1)}) > \mathcal{E}_{\text{GAT}}(\mathbf{X}^{(k)}), \quad (9)$$

*where $\mathcal{E}(\cdot)$ denotes the Dirichlet Energy, which quantifies the smoothness of node representations.*

The definition of Dirichlet Energy and the proof of Proposition 3.1 are provided in Appendix C.

**A Lightweight Alternative: Hop-Aware Attention.** To simplify the computation complexity ($O(|\mathcal{V}_p| \cdot d^2)$) of Hop-Diffused Attention while incorporating topology, we introduce learnable hop embeddings $\mathbf{h}_{\text{hop}}^{(h)} \in \mathbb{R}^d$ for each hop $h \in \{0, 1, \ldots, \tau\}$, where $\tau$ denotes the maximum hop count in the visual subgraphs. For visual embeddings $\mathbf{H}_P \in \mathbb{R}^{|\mathcal{V}_p| \times d}$, hop embeddings $\mathbf{H}_{\text{hop}}$ are added based on each node's hop information $h_v$:

$$\mathbf{H}_P \leftarrow \mathbf{H}_P + \mathbf{H}_{\text{hop}}, \quad \mathbf{H}_{\text{hop}} = [\mathbf{h}_{\text{hop}}^{(h_v)}]_{v \in \mathcal{V}_p}. \quad (10)$$

where $[\mathbf{h}_{\text{hop}}^{(h_v)}]_{v \in \mathcal{V}_p}$ represents the concatenation of learnable hop embeddings for each neighboring node in $\mathcal{V}_p$. Specifically, neighbors at the same hop distance (e.g., 1-hop neighbors) share the same hop embedding (e.g., $\mathbf{h}_{\text{hop}}^{(1)}$).

The above process is fully symmetric for textual embeddings, where hop embeddings are similarly added to the original textual features $\mathbf{H}_T$ to create topology-aware representations. This approach reduces the computational complexity to $O(|\mathcal{V}_p| \cdot d)$, while preserving critical hop information to guide the downstream model in adaptively learning the importance of the information from different hops.

### 3.3. Multi-Mapping QFormer

The most relevant work in multimodal graph learning for generative tasks, (Yoon et al., 2023), uses a simple linear projection (Tsimpoukelli et al., 2021) to map visual information into the language space but overlooks complex text-visual interactions. To address this, we propose the Multi-Mapping QFormer (MM-QFormer) framework, which takes the topology-integrated embeddings (e.g., Hop-Diffused $\mathbf{H}_P$ and $\mathbf{H}_T$) as input, inspired by (Li et al., 2023b). MM-QFormer employs learnable query tokens and a two-layer transformer, where shared self-attention facilitates text-query interactions, and cross-attention selectively extracts visual features relevant to textual inputs.

**Shared Self-Attention.** We initialize learnable query tokens as $\mathbf{Q}_v^{(0)} \in \mathbb{R}^{l_P \times d}$ for capturing intricate multi-modal information fusion, where $l_P = |\mathcal{V}_p| \times n_q$, and $n_q$ represents the number of multi-modal tokens per node. At each layer $t$, the query tokens from the previous layer $\mathbf{Q}_v^{(t-1)}$ are concatenated with the textual embeddings $\mathbf{H}_T$ to form a joint representation:

$$\mathbf{H}_{QT} = [\mathbf{Q}_v^{(t-1)}, \mathbf{H}_T] \in \mathbb{R}^{(l_P + |\mathcal{V}_t|) \times d}, \quad (11)$$

To make the learnable query can be condition on text information, the joint representation $\mathbf{H}_{QT}$ is then processed through shared self-attention:

$$\mathbf{H}'_{QT} = \text{SAT}[q = \mathbf{H}_{QT}, k = \mathbf{H}_{QT}, v = \mathbf{H}_{QT}], \quad (12)$$

where SAT($\cdot$) denotes multi-head self-attention. The updated query tokens are: $\mathbf{Q}'^{(t)}_v = \mathbf{H}'_{QT}[: l_P, :] \in \mathbb{R}^{l_P \times d}$. The query tokens are designed to integrate multimodal information by jointly learning from text and visual features. Through shared self-attention, the query tokens refine their representations while being conditioned on textual context, laying the foundation for better interacting with visual features in subsequent cross-modal operations.

**Modality Cross-Attention.** The cross-attention mechanism then aligns the updated query tokens $\mathbf{Q}'^{(t)}_v$ with these visual embeddings:

$$\mathbf{Q}^{(t)}_v = \text{CAT}[q = \mathbf{Q}'^{(t)}_v, k = \mathbf{H}_P, v = \mathbf{H}_P] \in \mathbb{R}^{l_P \times d}, \tag{13}$$

where CAT($\cdot$) denotes multi-head cross-attention. The cross-attention module enables the query tokens to extract relevant visual information by attending to the visual embeddings. This ensures that the multimodal query tokens are aligned with the most pertinent visual features while maintaining the contextual guidance established by the text.

**Feed-Forward Module.** Each cross-attention layer is followed by a feed-forward network (FFN) to further process the updated query tokens: $\mathbf{Q}^{(t)}_v = \text{FFN}(\mathbf{Q}^{(t)}_v)$, where FFN($\cdot$) is a two-layer fully connected network with a non-linear activation function in between, ensuring dimensional consistency: $\mathbf{Q}^{(t)}_v \in \mathbb{R}^{l_P \times d}$.

After $L$ layers of shared self-attention, cross-attention, and feed-forward modules, the final query tokens $\mathbf{Q}^{(L)}_v$ are projected into the multimodal semantic space, $\mathbf{f}(c_P(\mathcal{G}_p)) = \mathbf{Q}^{(L)}_v \in \mathbb{R}^{l_P \times d}$. Here, $\mathbf{f}(c_P(\mathcal{G}_p))$ represents the learned multimodal tokens, and $l_P = |\mathcal{V}_p| \times n_q$. Combined with textual prompts, these tokens form the input sequence for the downstream PLM, enabling it to effectively process open-ended tasks.

## 4. Experiments

### 4.1. Experiments Setups

**Datasets.** We evaluate our approach on two datasets that exhibit many-to-many text-image relationships. For the **generative task**, we use WIKIWEB2M (Burns et al., 2023), where the goal is to generate a section summary (i.e., first-sentence imputation) based on multimodal webpage content, including the page description, consecutive section text, images, and captions. For the **discriminative task**, we use ELE-FASHION (Zhu et al., 2024b), a product classification dataset where nodes represent products, edges capture co-purchase relationships, and text and images serve as attributes. To evaluate zero-shot node classification, we randomly hold out a set of unseen classes (5 out of 11 classes for the OPT-125M backbone and 9

for LLaMA-1B) for evaluation. Dataset details are provided in the Appendix F. Code is available at https://github.com/YennNing/Graph4MM.

**Baselines.** We evaluate PLMs, VLMs, and MMAG variants from MMGL (Yoon et al., 2023) under different input settings. For PLMs, we compare OPT-125M (Zhang et al., 2022b) and LLaMA-1B (Touvron et al., 2023), while for VLMs, we assess BLIP2-OPT-2.7B (Li et al., 2023b) and Qwen2-VL-7B-Instruct (Wang et al., 2024). The comparison includes four key settings: (1) using only a node's text (NODE'S TEXT), (2) adding image features (NODE'S TEXT & IMAGE), (3) incorporating subgraph-level text (SUBGRAPH'S TEXT), and (4) integrating both subgraph text and images (SUBGRAPH'S TEXT & IMAGE). For PLMs, we further explore fine-tuning and graph-enhanced modeling. Besides, in MMGL, a unique setting is SUBGRAPH'S T & I + GNN that utilizes GNNs to learn the subgraph's topological structure and incorporates it as a new modality into the visual and text input embeddings. Full baseline details are provided in the Appendix G.

### 4.2. Comparing Results

Our proposed Hop-Diffused and Hop-Aware MM-QFormer consistently outperforms pretrained VLMs, pretrained LLMs, and MMGL across both generative and discriminative tasks with on average 1.77% and 12.09% improvement resectively. From the experiment results, we find that pretrained VLMs perform the worst due to their pretraining focus on image captioning and simple QA, often generating irrelevant image descriptions instead of directly giving answers to the instructions. Among PLM-based methods, incorporating subgraph context consistently improves performance over using only node attributes, further validating the importance of modeling relevant multimodal contexts. MMGL performs well, but adding GCN embeddings degrades performance, likely due to the semantic gap between graph embeddings and LLM representations, which will be discussed in Section 4.3. Our MM-QFormer, with better text-image interaction modeling, achieves superior results compared to the strongest baselines. The introduction of Hop-Diffused Attention and Hop-Aware Attention further enhances MM-QFormer by modeling multi-hop connectivity, allowing the model to capture structure-aware multimodal information, thereby improving both generative and discriminative performance.

### 4.3. Discussions

**Ablation Studies.** The performance of MM-QFormer without structural modeling is reported in Table 4.1. We observe that incorporating MM-QFormer alone improves upon the baselines in most cases. Besides, we further conduct ablation experiments on both Hop-Diffused MM-QFormer

*Table 1.* Comparison of our approach with VLMs, PLMs, and MMGL variants on different backbones. In generative tasks, BLEU-4, ROUGE-L, and CIDEr measure language modeling accuracy (higher is better). In discriminative tasks, ROUGE-L (R-L) evaluates response accuracy, while Accuracy (Acc), Recall (Rec), and Precision (Pre) assess classification correctness. The best results are **bolded**, while the best baseline results are underlined.

| Backbone | Method | Generative Task | | | Discriminative Task | | | |
|---|---|---|---|---|---|---|---|---|
| | | BLEU-4 | ROUGE-L | CIDEr | R-L | Acc (%) | Rec (%) | Pre (%) |
| VLMs — BLIP | NODE'S TEXT (T) | 0.0000 | 0.0496 | 0.0060 | 0.1450 | 21.89 | 9.62 | 16.27 |
| | SUBGRAPH'S TEXT | 0.0000 | 0.0530 | 0.0063 | 0.1907 | 31.37 | 14.49 | 31.73 |
| VLMs — QwenVL | NODE'S TEXT | 0.0000 | 0.1223 | 0.0069 | 0.0793 | 11.00 | 4.43 | 10.90 |
| | SUBGRAPH'S TEXT | 0.0000 | 0.1192 | 0.0084 | 0.1233 | 12.33 | 5.78 | 14.26 |
| OPT-125M — PLM | NODE'S TEXT | 0.0078 | 0.1871 | 0.0783 | 0.2270 | 68.57 | 40.44 | 48.76 |
| | SUBGRAPH'S TEXT | 0.0164 | 0.2198 | 0.1551 | 0.2762 | 67.68 | 33.14 | 40.26 |
| OPT-125M — MMGL | NODE'S TEXT | 0.0642 | 0.3807 | 0.6241 | 0.7047 | 70.33 | 66.67 | 66.67 |
| | SUBGRAPH'S TEXT | 0.0770 | 0.3992 | 0.7606 | 0.8149 | 99.74 | 71.03 | 71.43 |
| | NODE'S TEXT & IMAGE (I) | 0.0643 | 0.3825 | 0.6371 | 0.7180 | 73.90 | 68.67 | 83.33 |
| | SUBGRAPH'S TEXT & IMAGE | 0.0778 | 0.4041 | 0.7712 | 0.8144 | 99.85 | 83.25 | 83.33 |
| | SUBGRAPH'S T & I + GNN | 0.0633 | 0.3814 | 0.6326 | 0.5771 | 70.89 | 56.08 | 62.50 |
| OPT-125M — Graph4MM (Ours) | MM-QFORMER | 0.0769 | 0.4044 | 0.7684 | 0.8195 | 100.00 | 100.00 | 100.00 |
| | HOP-AWARE MM-QFORMER | **0.0801** | 0.4063 | 0.7736 | 0.8097 | 100.00 | 100.00 | 100.00 |
| | HOP-DIFFUSED MM-QFORMER | 0.0800 | **0.4076** | **0.7831** | **0.8282** | **100.00** | **100.00** | **100.00** |
| Llama-1B — PLM | NODE'S TEXT | 0.0757 | 0.3691 | 0.7185 | 0.7081 | 94.76 | 78.07 | 83.11 |
| | SUBGRAPH'S TEXT | 0.0960 | 0.3949 | 0.9179 | 0.7327 | 99.79 | 81.42 | 81.27 |
| Llama-1B — MMGL | NODE'S TEXT | 0.0947 | 0.4375 | 0.9046 | 0.7083 | 94.80 | 78.12 | 83.19 |
| | SUBGRAPH'S TEXT | 0.1087 | 0.4426 | 1.0426 | 0.7375 | 99.84 | 81.51 | 81.58 |
| | NODE'S TEXT & IMAGE | 0.0981 | 0.4449 | 0.9367 | 0.6696 | 97.26 | 84.33 | 88.38 |
| | SUBGRAPH'S TEXT & IMAGE | 0.1157 | 0.4685 | 1.1072 | 0.7446 | 98.07 | 84.55 | 85.50 |
| | SUBGRAPH'S T & I + GNN | 0.1003 | 0.4487 | 0.9631 | 0.3823 | 86.73 | 62.80 | 61.44 |
| Llama-1B — Graph4MM (Ours) | MM-QFORMER | 0.1168 | 0.4699 | 1.1108 | 0.8321 | 99.88 | 99.86 | 99.87 |
| | HOP-AWARE MM-QFORMER | **0.1186** | **0.4731** | **1.1262** | **0.8891** | 99.87 | 89.86 | 89.88 |
| | HOP-DIFFUSED MM-QFORMER | 0.1177 | 0.4713 | 1.1221 | 0.8822 | **100.00** | **100.00** | **100.00** |

and Hop-Aware MM-QFormer by individually removing structural priors from the image and text modalities. The result is shown in Table 4.3. In all cases, removing structural information leads to a performance drop, reinforcing the importance of multi-hop structural information in guiding multimodal learning. Notably, removing the structural information in the image input leads to a severer performance drop, which is because text can provide structural information through explicitly added some descriptive hints (e.g., "context from 1-hop neighbor") in the prompt. Such prompts, illustrated in Appendix Q, allow the language modality to retain partial access to structural information. In contrast, without structural information, images from different hops are treated with the same level of importance, leading to suboptimal performance.

*Table 2.* Ablation study on Hop-Diffused and Hop-Aware Attention in the generative setting with OPT-125M. The first row shows original results, while $-t_{\mathcal{G}'}$ and $-p_{\mathcal{G}'}$ indicate the removal of graph adjacency heuristics for text and image subgraphs.

| Ablation Variant | BLEU-4 | ROUGE-L | CIDEr |
|---|---|---|---|
| Hop-Diffused MM-QFormer | 0.0800 | 0.4076 | 0.7831 |
| - $t_{\mathcal{G}'}$ Hop-Diffused Attention | 0.0786 | 0.4065 | 0.7765 |
| - $p_{\mathcal{G}'}$ Hop-Diffused Attention | 0.0769 | 0.4044 | 0.7684 |
| Hop-Aware MM-QFormer | 0.0801 | 0.4063 | 0.7736 |
| - $t_{\mathcal{G}'}$ Hop-Aware Attention | 0.7943 | 0.4047 | 0.7736 |
| - $p_{\mathcal{G}'}$ Hop-Aware Attention | 0.0769 | 0.4044 | 0.7684 |

**Robustness from Hop-diffused to Hop-aware Attention.** Hop-Aware Attention reduces computational complexity while maintaining strong performance. Replacing either the image or text modality with Hop-Diffused Attention across four datasets (results shownin Table 3) shows that even lightweight structural priors ensure stable performance, and hybrid approaches may sometimes outperform full Hop-Diffused Attention, due to the complementary inductive biases introduced by modality-specific structural modeling.

*Table 3.* Comparison of replacing text or vision structural priors in Hop-Diffused MM-QFormer with Hop-Aware Attention on the generative task using OPT-125M and LLaMA-1B. Best results are in red, second-best in yellow, and third-best in grey. $t_{\mathcal{G}'}$ and $p_{\mathcal{G}'}$ denote text and visual subgraphs in MM-QFormer, respectively.

| Model | Method | BLEU-4 | ROUGE-L | CIDEr |
|---|---|---|---|---|
| OPT-125M | Hop-Diffused (HD) | 0.0800 | 0.4076 | 0.7831 |
| | Hop-Aware (HA) | 0.0801 | 0.4063 | 0.7736 |
| | $t_{\mathcal{G}'}$-HA + $p_{\mathcal{G}'}$-HD | 0.0822 | 0.4094 | 0.7947 |
| | $t_{\mathcal{G}'}$-HD + $p_{\mathcal{G}'}$-HA | 0.0802 | 0.4063 | 0.7826 |
| LLaMA-1B | Hop-Diffused (HD) | 0.1177 | 0.4713 | 1.1221 |
| | Hop-Aware (HA) | 0.1186 | 0.4731 | 1.1262 |
| | $t_{\mathcal{G}'}$-HA + $p_{\mathcal{G}'}$-HD | 0.1176 | 0.4709 | 1.1199 |
| | $t_{\mathcal{G}'}$-HD + $p_{\mathcal{G}'}$-HA | 0.1179 | 0.4744 | 1.1207 |

**Should Graph be Encoded as a Modality?**

A key question in multi-model graph learning is whether

graph structure should be explicitly encoded as an additional modality. MMGL adopts a straightforward approach by using a GCN to encode $\mathcal{G}$, then directly adding its graph representations to the pretrained vision and language embeddings before feeding them into the LLM. The GCN itself is trained jointly with the downstream task loss. However, as shown in the first two rows of Table 4.3, this method provides little to no improvement and often performs worse than using only raw text and vision attributes. To further investigate this, we experimented with treating graph topology as an independent modality in the generative setting with OPT-125M. We explored two designs: (1) projecting GCN-learned node embeddings into global graph tokens to represent overall graph structure, and (2) mapping them into node tokens that encode local topology at the node level. Despite these efforts, neither approach yielded meaningful performance gains. A fundamental reason for this failure is the semantic gap between graph-derived topology embeddings and pretrained vision-language representations. Unlike large-scale pretrained models, which align text and image representations through extensive multimodal training, GCNs operate on small, sparsely labeled subgraphs, limiting their expressiveness. To formalize this gap, we propose the following proposition:

**Proposition 4.1.** *Let $\mathcal{G}$ be a small-scale graph data distribution, $G \sim r(\mathcal{G})$, and let $\mathcal{X}$ be a large-scale data distribution (e.g., text or image), $X \sim p(\mathcal{X})$. Suppose $f_\theta^{(G)}$ is a GCN-based encoder trained only on $\mathcal{G}$ to produce representations $\mathbf{z}_G = f_\theta^{(G)}(G)$, while $f_\phi^{(X)}$ is a large-scale pre-trained encoder (e.g., LLM or ViT) producing $\mathbf{z}_X = f_\phi^{(X)}(X)$. Then, for almost all samples $x \in \mathcal{X}$, the mutual information satisfies: $I(\mathbf{z}_G; X) \ll I(\mathbf{z}_X; X)$, implying a fundamental gap in expressive power between $\mathbf{z}_G$ and $\mathbf{z}_X$.*

The proof is in Appendix D. Additionally, pretrained vision and language models benefit from strong cross-modal alignment, allowing text and image representations to be semantically consistent. In contrast, graph embeddings lack such alignment, further widening the gap. Thus, we propose use graph structure as a guide for selecting and fusing multimodal data rather than treating it as a separate modality.

*Table 4.* Comparison of different methods for incorporating topology in the generative setting using the OPT-125M backbone.

| Method | BLEU-4 | ROUGE-L | CIDEr |
|---|---|---|---|
| Subgraph's T & I (Emb) | 0.0649 | 0.3800 | 0.6370 |
| + Graph Embeddings | 0.0633 | 0.3814 | 0.6326 |
| Subgraph's T& I | 0.0788 | 0.4051 | 0.7790 |
| + Graph Tokens | 0.0796 | 0.4052 | 0.7736 |
| + Node Tokens | 0.0782 | 0.4045 | 0.7651 |
| Ours (Hop-Diffused) | 0.0800 | 0.4076 | 0.7831 |

## 5. Related Work

**Vision-Language Models** . Recent vision-language models leverage frozen pretrained components, either fixing the image encoder (Chen et al., 2020; Li et al., 2020; Zhang et al.; Zhai et al., 2022) or freezing the LLM to utilize its knowledge for vision-to-language generation (Tsimpoukelli et al., 2021; Alayrac et al., 2022b; Chen et al., 2022a). When using a frozen LLM (Touvron et al., 2023; Zhang et al., 2022b), the key challenge is aligning visual features to the text space. Frozen (Tsimpoukelli et al., 2021) finetunes an image encoder as soft prompts for an LLM, while Flamingo (Alayrac et al., 2022b) injects visual features via trainable cross-attention layers. BLIP-2 (Li et al., 2023b) introduces learnable query-based transformers to project visual features into the LLM's space. Recent large-scale VLMs (Wang et al., 2024) employ multi-stage pretraining and multi-task finetuning to further enhance vision-language understanding. However, most VLMs still focus on one-to-one image-text alignment or naive concatenation of multimodal tokens for multi-image scenarios, failing to capture complex many-to-many relationships where multiple images correspond to different textual fragments. We address this by extending multimodal learning to graph structures, enabling structured cross-modal interactions.

**Graph Neural Networks for Multimodal Data.** Heterogeneous Graph Neural Networks (HGNNs) extend standard GNNs (Veličković et al., 2018; Wu et al., 2020) to process multimodal graphs by integrating heterogeneous node attributes. This is typically done by extracting embeddings with frozen encoders and fusing modalities at different levels: input (Schlichtkrull et al., 2018; Zhang et al., 2019), intermediate (Hu et al., 2020), or final representation layers (Yoon et al., 2022). While effective for node classification and link prediction, HGNNs are not well suited for language generation or open-ended reasoning. More discussion on Multi-modal Knowledge Graph (MMKG) can be seen in Appendix H. MMGL (Yoon et al., 2023) introduced multimodal graph learning for text generation by combining textual and visual encoders with GNNs. However, aligning graph-based representations with large language models remains challenging. We address this by graph-heuristic multimodal fusion mechanism to enhance many-to-many vision-language understanding.

**Learning on the Text-Attributed Graphs.** Integrating structured graph data with unstructured text has evolved from early MLP- and transformer-based fusion methods (Feng et al., 2020; Zhang et al., 2022c; Lin et al., 2019), which struggled to capture complex contextual dependencies and were insufficient for language-centric tasks like open-ended QA. Recent efforts explore LLMs for graph learning (He et al., 2023; Zhu et al., 2024c; Lin et al., 2024),

primarily refining graph representations within the LLM framework. GraphTranslator (Zhang et al., 2024b) aligns graph embeddings with LLMs using textual descriptions, but its dependence on explicit, comprehensive language annotations of graph structures limits its applicability. Our approach extends multimodal graph learning by integrating vision, enabling structured multimodal fusion for both language generation and graph-based reasoning.

## 6. Conclusion

In this paper, we propose representing the complex interactions between different modalities using a multimodal attributed graph and introduce a theoretically guided, adjacency-aware multimodal fusion framework, Hop-Diffused and Hop-Aware MM-QFormer. Our approach effectively enhances performance in both generative and discriminative tasks within multimodal learning. We also discuss whether graph structure should be treated as a separate modality in the era of foundation models.

## Acknowledgements

We would like to thank Ke Jiang and Xiao Lin from the University of Illinois Urbana-Champaign for their valuable suggestions to improve our work, as well as Jiaxuan You for the insightful discussions during the CS598: Deep Learning with Graphs course. This work is supported by National Science Foundation under Award No. IIS-2117902. The views and conclusions are those of the authors and should not be interpreted as representing the official policies of the funding agencies or the government.

## Impact Statement

This paper presents work whose goal is to advance the field of Machine Learning. There are many potential societal consequences of our work, none which we feel must be specifically highlighted here.

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

# Appendix Content

## A. $\mathcal{A}$ Remains the Property of a Valid Attention Matrix

**Proposition A.1.** *The diffused attention matrix $\mathcal{A} = \sum_{i=0}^{\infty} \theta_i A^i$ where $A^i$ is the $i$-th power of the base attention matrix $A$, and $\theta_i$ is a set of positive weights satisfying $\sum_{i=0}^{\infty} \theta_i = 1$, remains a valid attention matrix with each row summing to 1, i.e., $\sum_j \mathcal{A}_{ij} = 1, \forall i$.*

*Proof.* Since $A$ is obtained by row-wise softmax normalization, it satisfies $\sum_j A_{ij} = 1$. Since matrix multiplication preserves row-wise summation in stochastic matrices, any power $A^i$ satisfies $\sum_j (A^i)_{ij} = 1$ for all $i \geq 0$. The weight coefficients $\theta_i$ form a probability distribution with $\sum_{i=0}^{\infty} \theta_i = 1$, ensuring that when computing row sums in $\mathcal{A}$, we have:

$$\sum_j \mathcal{A}_{ij} = \sum_j \sum_{i=0}^{\infty} \theta_i (A^i)_{ij} = \sum_{i=0}^{\infty} \theta_i \sum_j (A^i)_{ij} = \sum_{i=0}^{\infty} \theta_i \cdot 1 = 1. \tag{14}$$

Thus, $\mathcal{A}$ is row-stochastic, meaning each row sums to 1. This confirms that the hop-diffused attention mechanism essentially adjusts the attention scores based on global connectivity at different hop distances, maintaining the fundamental property of an attention matrix. $\qquad\square$

## B. Global Connectivity Aware by Hop-diffused Attention

**Proposition B.1.** *Hop-Diffused Attention is a generalized form of Personalized PageRank (PPR) with an adaptive multi-hop weighting scheme. Given an attention transition matrix $A_{att}$, its diffusion process follows:*

$$\mathcal{A} = \sum_{i=0}^{\infty} \theta_i A_{att}^i, \quad \sum_{i=0}^{\infty} \theta_i = 1, \quad \theta_i > 0. \tag{15}$$

*For $\theta_i = \alpha(1-\alpha)^i$, this recovers the standard PPR formulation.*

*Proof.* Personalized PageRank is defined as:

$$A_{\text{ppr}} = \alpha(I - (1-\alpha)A_{\text{att}})^{-1}. \tag{16}$$

Expanding the inverse as a Neumann series, we obtain:

$$A_{\text{ppr}} = \alpha \sum_{i=0}^{\infty} (1-\alpha)^i A_{\text{att}}^i. \tag{17}$$

Compared with the Hop-Diffused Attention formulation,

$$\mathcal{A} = \sum_{i=0}^{\infty} \theta_i A_{\text{att}}^i, \tag{18}$$

where $\theta_i$ is a probability distribution with $\sum_{i=0}^{\infty} \theta_i = 1$, we see that PPR is a special case where $\theta_i = \alpha(1-\alpha)^i$. This proposition further illustrates that akin to personalized PageRank, Hop-Diffused Attention preserves the global graph structure while adaptively regulating the influence of multi-hop neighbors, where the impact of more distant neighbors decays exponentially. $\qquad\square$

## C. Hop-Diffused Attention is Less Prone to Over-Smoothing than GAT

### C.1. Measuring Over-Smoothing: Dirichlet Energy

**Definition C.1** (Dirichlet Energy). Let $\mathcal{G} = (\mathcal{V}, \mathcal{E})$ be a graph with node set $\mathcal{V}$ and edge set $\mathcal{E}$. Let $X^n \in \mathbb{R}^{|\mathcal{V}| \times m}$ denote the node feature matrix at the $n$-th layer of a GNN. The Dirichlet energy, which quantifies the smoothness of node representations, is defined as:

$$\mathcal{E}(X^n) = \frac{1}{|\mathcal{V}|} \sum_{i \in \mathcal{V}} \sum_{j \in \mathcal{N}_i} \|X_i^n - X_j^n\|_2^2, \tag{19}$$

where $\mathcal{N}_i$ denotes the neighborhood of node $i$. If a GNN exhibits over-smoothing, then node representations converge to similar values across the graph, implying that $\mathcal{E}(X^n) \to 0$.

### C.2. Theoretical Analysis

**Proposition C.2.** *Let $X^{(1)}$ be the node representation matrix after the Hop-Diffused Attention module, and $X^{(k)}$ be the representation at the $k$th layer of GAT. When both methods aggregate information from $k$-hop neighbors, Hop-Diffused Attention retains higher Dirichlet energy, preserving more feature variance and mitigating over-smoothing. Formally, for large $k$,*

$$\mathcal{E}_{\text{Hop-Diffused}}(X^{(1)}) > \mathcal{E}_{GAT}(X^{(k)}). \tag{20}$$

*Proof.* In GAT, the updated rule for the $l^{\text{th}}$ graph attention layer can be written as:

$$X^{(l+1)} = \sigma(A_{\text{att}}^{(l)} X^{(l)} W^{(l)}), \tag{21}$$

where $A_{\text{att}}$ is the learned aggregation matrix, $W^{(l)}$ is a trainable transformation matrix, and $\sigma$ is a point-wise nonlinear activation function. Expanding recursively for $k$ layers, we obtain:

$$X^{(k)} = \sigma(A_{\text{att}}^{(k)} \sigma(A_{\text{att}}^{(k-1)} \cdots \sigma(A_{\text{att}}^{(0)} X^{(0)} W^{(0)}) \cdots W^{(k-1)}) W^{(k)}). \tag{22}$$

To handle the nonlinearity of $\sigma$, we assume the activation function satisfies $0 \leq \frac{\sigma(x)}{x} \leq 1$ for $x \neq 0$, and $\sigma(0) = \sigma'(0)$ or 1 if $\sigma'(0)$ is undefined, refering to Wu et al. (2023). This holds for common activations such as ReLU and LeakyReLU. Under this assumption, any activation $\sigma(y)$ on vector $y$ can be rewritten as:

$$\sigma(y) = \text{diag}\left(\frac{\sigma(y)}{y}\right) y. \tag{23}$$

We denote $D_i^{(k)}$ as the diagonal transformation matrix induced by the activation at layer $k$, satisfying

$$\text{diag}(0) \preceq D_i^{(k)} \preceq \text{diag}(1). \tag{24}$$

The representation $X_i^{(k+1)}$ can be recursively expanded as:

$$X_{\cdot i}^{(k+1)} = \sum_{\substack{j_{k+1}=i \\ (j_k,\ldots,j_0) \in [d]^{k+1}}} \left(\prod_{l=0}^{k} W_{j_l j_{l+1}}^{(l)}\right) D_{j_{k+1}}^{(k)} A^{(k)} \cdots D_{j_1}^{(0)} A^{(0)} X_{\cdot j_0}^{(0)}. \tag{25}$$

Here, $A^{(k)}$ is a row-stochastic aggregation matrix. Since all $D^{(k)}$ and $A^{(k)}$ have spectral norm less than or equal to 1, and each weight matrix $W^{(k)}$ is bounded, it follows that:

$$\left\| X^{(k)} \right\| \leq c^k \cdot \left\| X^{(0)} \right\| \tag{26}$$

for some constant $0 < c < 1$. The Dirichlet energy is given by:

$$\mathcal{E}_{\text{GAT}}(X^{(k)}) = \frac{1}{n} \text{Tr}\left(X^{(k)\top} L X^{(k)}\right), \tag{27}$$

Let $\lambda_{\max}$ be the largest eigenvalue of the Laplacian $L$ for $X^{(k)}$,

$$\text{Tr}(Z^\top L Z) \leq \lambda_{\max} \cdot \text{Tr}(Z^\top Z) = \lambda_{\max} \cdot \|Z\|_F^2. \tag{28}$$

Applying this to Eq. (27):

$$\mathcal{E}_{\text{GAT}}(X^{(k)}) \leq \frac{\lambda_{\max}}{n} \cdot \|X^{(k)}\|_F^2. \tag{29}$$

Because $\|X^{(k)}\|_F \leq \sqrt{r} \|X^{(k)}\| \leq \sqrt{r} \, c^k \|X^{(0)}\|$ with $r = \text{rank}(X^{(k)}) \leq d$, we have

$$\mathcal{E}_{\text{GAT}}(X^{(k)}) \leq \frac{\lambda_{\max}}{n} \, r \, c^{2k} \, \|X^{(0)}\|^2. \tag{30}$$

Letting $\gamma = c^2 \in (0, 1)$ and absorbing constants, we arrive at:

$$\mathcal{E}_{\text{GAT}}(X^{(k)}) = O(\gamma^k \cdot \mathcal{E}(X^{(0)})), \tag{31}$$

indicating exponential decay of Dirichlet energy. Hence, as $k \to \infty$, the energy tends to zero:

$$\lim_{k \to \infty} \mathcal{E}_{\text{GAT}}(X^{(k)}) = 0, \tag{32}$$

which reflects the over-smoothing effect in deep GATs.

**Hop-Diffused Attention Analysis**  For Hop-Diffused Attention, we use a weighted sum of attention powers:

$$\mathcal{A} = \sum_{i=0}^{K} \theta_i A_{\text{att}}^i. \tag{33}$$

This modifies the feature propagation as:

$$X^{(1)} = \mathcal{A} X^{(0)} W. \tag{34}$$

Applying Dirichlet Energy to the Hop-Diffused Attention Propagation:

$$\mathcal{E}_{\text{Hop-Diffused}}(X^1) = \frac{1}{|\mathcal{V}|} \sum_{u \in \mathcal{V}} \sum_{v \in \mathcal{N}_u} \left\| \sum_{i=0}^{K} \theta_i A_{\text{att}}^i X W_u - \sum_{i=0}^{K} \theta_i A_{\text{att}}^i X W_v \right\|^2. \tag{35}$$

Using Jensen's inequality and the convexity of the squared norm, we approximate:

$$\mathcal{E}_{\text{Hop-Diffused}}(X^1) \leq \sum_{i=0}^{K} \theta_i \mathcal{E}(A_{\text{att}}^i X). \tag{36}$$

Using the eigenvalue decomposition of $A_{\text{att}}$, we approximate:

$$\mathcal{E}(A_{\text{att}}^i X) = \lambda_{\max}^i \mathcal{E}(X). \tag{37}$$

Thus, the energy formulation simplifies to:

$$\mathcal{E}_{\text{Hop-Diffused}}(X^1) = \sum_{i=0}^{K} \theta_i \lambda_{\max}^i \mathcal{E}(X). \tag{38}$$

Since $\sum_{i=0}^{K} \theta_i \lambda_{\max}^i$ remains larger than $\gamma^k$ for sufficiently large $k$, it follows that:

$$\mathcal{E}_{\text{Hop-Diffused}}(X^1) > \mathcal{E}_{\text{GAT}}(X^k). \tag{39}$$

Thus, Hop-Diffused Attention retains higher feature heterogeneity, making it less prone to over-smoothing compared to directly applying GAT. We also provide a simulation study comparing Dirichlet energy under small values of $K$ in Appendix L. □

# D. Mutual Information Gap Between Small-Scale Graph Encoders and Large-Scale Pretrained Models

**Proposition D.1.** *Let $\mathcal{G}$ be a small-scale graph data distribution, $G \sim r(\mathcal{G})$, and let $\mathcal{X}$ be a large-scale data distribution (e.g., text or image), $X \sim p(\mathcal{X})$. Suppose $f_\theta^{(G)}$ is a GCN-based encoder trained* only *on $\mathcal{G}$ to produce representations $\mathbf{z}_G = f_\theta^{(G)}(G)$, while $f_\phi^{(X)}$ is a large-scale pre-trained encoder (e.g., LLM or ViT) producing $\mathbf{z}_X = f_\phi^{(X)}(X)$. Then, for almost all samples $x \in \mathcal{X}$, the mutual information satisfies*

$$I(\mathbf{z}_G; X) \ll I(\mathbf{z}_X; X), \tag{40}$$

*which implies a fundamental gap in expressive power between $\mathbf{z}_G$ and $\mathbf{z}_X$.*

*Proof.* Let $G$ denote graph-structured inputs drawn from a small-scale distribution $r(\mathcal{G})$, and let $X$ denote data (text, images, etc.) from a large-scale distribution $p(\mathcal{X})$. We use $\mathbf{z}_G = f_\theta^{(G)}(G)$ to denote the representation learned by a GCN on $\mathcal{G}$, and $\mathbf{z}_X = f_\phi^{(X)}(X)$ to denote the representation learned by a large-scale pre-trained model on $\mathcal{X}$.

Let $\mathbf{z}_G$ have dimension $d$. By Shannon's definition of mutual information,

$$I(\mathbf{z}_G; G) = H(\mathbf{z}_G) - H(\mathbf{z}_G \mid G). \tag{41}$$

For a deterministic encoder $f_\theta^{(G)}$, the conditional entropy $H(\mathbf{z}_G \mid G)$ is small, so

$$I(\mathbf{z}_G; G) \approx H(\mathbf{z}_G). \tag{42}$$

Since the dataset $\mathcal{D}_G$ is small, we have

$$H(\mathbf{z}_G) \leq \log|\mathcal{D}_G|. \tag{43}$$

Thus,

$$I(\mathbf{z}_G; G) \leq \log|\mathcal{D}_G|, \tag{44}$$

implying a strict upper bound on the captured information.

Consider $\mathbf{z}_X = f_\phi^{(X)}(X)$ obtained via pre-training on $\mathcal{X}$. With large data coverage $|\mathcal{D}_X|$ and a sufficiently expressive model, the entropy $H(\mathbf{z}_X)$ can be much higher:

$$H(\mathbf{z}_X) \approx \log|\mathcal{D}_X| \quad (\text{where } |\mathcal{D}_X| \gg |\mathcal{D}_G|). \tag{45}$$

Hence,

$$I(\mathbf{z}_X; X) = H(\mathbf{z}_X) - H(\mathbf{z}_X \mid X) \approx \log|\mathcal{D}_X|, \tag{46}$$

which is significantly larger than $\log|\mathcal{D}_G|$ in most practical scenarios.

When aligning $\mathbf{z}_G$ and $\mathbf{z}_X$ in a shared semantic space, note that

$$I(\mathbf{z}_G; X) \leq I(\mathbf{z}_G; G) + I(G; X). \tag{47}$$

Since $I(\mathbf{z}_G; G) \leq \log|\mathcal{D}_G|$ and $I(G; X)$ is typically small (the graph domain is narrow or quite distinct from $\mathcal{X}$), it follows that

$$I(\mathbf{z}_G; X) \ll I(\mathbf{z}_X; X). \tag{48}$$

The above inequalities establish that representations $\mathbf{z}_G$ learned *only* on a small-scale graph dataset $\mathcal{G}$ generally exhibit far lower mutual information with broader data $X \in \mathcal{X}$. This accounts for the substantial expressiveness gap and difficulty in aligning $\mathbf{z}_G$ with $\mathbf{z}_X$ in a common semantic space. $\square$

# E. Graph Modeling

**Defining Multi-modal Data Connectivity.** In real-world scenarios, multimodal data often interact in intricate ways. For instance, in document understanding, integrating both text (e.g., section content or page titles) and images (Yasunaga et al., 2022) enables language models to better comprehend the context and improve QA performance. In e-commerce, user interactions with items, represented as text descriptions and images, often reveal implicit correlations that (Wei et al., 2024), when effectively modeled, enhance recommendation accuracy. Similarly, in biomedicine, understanding interactions between molecular compositions of proteins and their corresponding visual structures can aid in predicting new protein attributes. The challenge lies in organizing and preserving these complex relationships between multimodal data, which are often interdependent and hierarchical. To address this, we propose a graph-based framework that captures these connections explicitly. In the following sections, we outline two key approaches for defining and identifying such multimodal relationships:

(a) When multimodal information exhibits intuitive connectivity in its original form, we leverage this natural structure. For instance, in multimodal documents, closer elements (e.g., adjacent paragraphs or text and images within the same section) are more relevant. Accordingly, we construct content graphs based on paragraph relationships, text-image links, and image-caption associations.

(b) When no predefined connectivity exists, relevance is task-specific. For example, in e-commerce platforms, graphs are constructed from user co-click patterns, where frequently co-clicked items indicate higher relevance. Each item is associated with textual descriptions and corresponding images.

# F. Dataset Details

### F.1. WikiWeb2M

WikiWeb2M (Burns et al., 2023) is a multimodal webpage dataset extending WIT (Srinivasan et al., 2021) by incorporating full-page structural metadata, section text, images, captions, and hierarchical relationships. We adopt the section summarization task, where the model generates a missing first sentence for a section given its text, images, and multimodal context from related sections on the same page, which follows the data processing process in MMGL (Yoon et al., 2023). Each section and the overall page description are represented as nodes, with text and images as attributes. Edges capture the relationships between consecutive sections, links between sections and their images and captions, caption-to-section connections, and links between page descriptions and each section. Due to storage constraints, we randomly sample 10K Wikipedia pages, resulting in 13,539 section summary samples for training and 1,768 for testing.

### F.2. Ele-Fashion

Ele-Fashion (Zhu et al., 2024b) is a multimodal node classification dataset constructed from Amazon-Fashion (Ni et al., 2019), designed for evaluating graph-aware multimodal learning. Each node represents a product, with textual and visual attributes derived from product titles and images. Edges encode co-purchase relationships between products. The classification task aims to predict product categories using both multimodal attributes and graph structure. Ele-Fashion consists of 11 product categories, 97,766 nodes, and 199,602 edges, with an average node degree of 4.08 and an edge homophily score of 0.7675. Each node has a textual description, and most nodes are also associated with an image. For the experiments with OPT-125M backbone, we held out 5 unseen classes for evaluation: {3,4,7,9,11}, and due to the strong reasoning ability of LLaMA-1B, we enhance the task difficulty by holding out 9 unseen classes for evaluation: {0, 9, 6, 10, 1, 2, 3, 11, 7}.

# G. Detailed Baselines

We compare various transformer-based pretrained language models (PLMs), vision-language models (VLMs), and MMAG learning variants from MMGL (Yoon et al., 2023).

For PLMs, we evaluate two backbones: (a) OPT-125M (Zhang et al., 2022b), as used in MMGL; (b) LLaMA-1B (Touvron et al., 2023), a 1B-parameter model from the latest LLaMA 3.2 family, chosen as a computationally feasible representative of large language models.

For VLMs, we compare (a) BLIP2-OPT-2.7B (Li et al., 2023b), which employs query-based vision-language alignment

but lacks structured multimodal context modeling, and (b) Qwen2-VL-7B-Instruct (Wang et al., 2024), a state-of-the-art open-source VLM supporting multi-image input.

For each PLM backbone, we evaluate:

- NODE'S TEXT (T): Input single-node text attributes to a frozen PLM.

- SUBGRAPH'S T: Input text attributes of node itself and of other nodes in $\mathcal{G}_t$.

For VLM backbones, we compare inference performance on the same two settings as PLM. Fine-tuned VLMs are excluded due to the significant scale difference from LLMs.

For MMGL(Yoon et al., 2023), we evaluate its various proposed settings:

- NODE'S TEXT: Fine-tune the PLM using single-node text attributes with parameter efficient tuning (PEFT), including prefix tuning (Li & Liang, 2021) or lora (Hu et al., 2021).

- SUBGRAPH'S TEXT: Fine-tune the PLM with text attributes of subgraph nodes.

- NODE'S TEXT & IMAGE (I): Input text and image attributes of a single node. Visual features are mapped to tokens via a frozen encoder and linear projection. The PLM is fine-tuned with PEFT.

- SUBGRAPH'S TEXT & IMAGE: Extend Node's T & I to include text and image attributes of neighboring nodes in $\mathcal{G}_t$ and $\mathcal{G}_p$.

- SUBGRAPH'S T& I + GNN: Map text and visual features to embeddings, process $\mathcal{G}$ with a GCN, and add graph embeddings to the PLM's token embeddings.

## H. Related Work about Multi-Modal Knowledge Graphs

Multimodal Knowledge Graphs (MMKG) integrate structured relational triples (Li et al., 2025b; 2023d; Zeng et al., 2023a) in graph learning (Fu & He, 2021; Fu et al., 2022; Xu et al., 2023; Zeng et al., 2023b; Zheng et al., 2024; Li et al., 2024; Wei et al., 2022b; Zhu et al., 2024a; Li et al., 2023a; Fu et al., 2024; Xu et al., 2024c; Wang et al., 2025; Li et al., 2025a) with auxiliary modalities such as images, text, or numerical attributes (Chen et al., 2024). Early efforts focused on MMKG dataset construction (Zhu et al., 2022), enriching entities in traditional KGs (e.g., Freebase, DBpedia (Ferrada et al., 2017)) with visual features to support tasks like link prediction (Ban et al., 2024; Tieu et al., 2025) or entity matching (Li et al., 2023e). Recent work expands to multimodal knowledge graph completion (Chen et al., 2022b; Xu et al., 2022), cross-modal reasoning (Zheng et al., 2023; Zhang et al., 2022a), and adaptive entity representation via modality-aware fusion (Zhang et al., 2024c). For example, MOMOK (Zhang et al., 2024c) proposes a mixture-of-experts model (Ai et al., 2023) that dynamically attends to different modalities during embedding learning, while MarT (Zhang et al., 2022a) tackles analogical reasoning by aligning multimodal relational patterns. These methods largely operate at the entity or triple level, using contrastive or attention-based fusion (Xu et al., 2024b) to enhance representation learning, thus lacking the capacity to support both generative (Xu et al., 2024a; Zhang et al., 2025; Wei et al., 2024; Ning et al., 2024) and discriminative tasks simultaneously.

## I. Implementation Details

**Subgraph Extraction.** Different strategies are used based on graph density. In WikiWeb2M, where edges and images are sparse, we follow MMGL's approach, selecting 10 textual neighbors and 4 image neighbors, along with the target node's attributes. In contrast, Ele-Fashion has a denser structure with abundant edges. To balance structure retention and complexity, we keep all first-hop neighbors and randomly sample two second-hop neighbors, capping textual and visual subgraphs at 11 nodes (target + 10 neighbors).

**PLM Input Length.** For single-node input, we set 512 tokens for generative tasks and 128 tokens for discriminative tasks. When including subgraph context, limits increase to 1024 tokens (generative) and 512 tokens (discriminative) to accommodate richer information.

**Parameter-Efficient Tuning.** We apply Prefix-Tuning for OPT-125M and LoRA for LLaMA-1B, optimizing adaptability while ensuring efficient updates. The hyper-parameters of PEFT, including the number of prefix tokens (20) and the row-rank(64) of lora, keeps the same for all baselines and our method for fair comparison. The vision encoder is CLIP. All experiments were conducted on computing nodes equipped with 2 NVIDIA A100 or 2 NVIDIA Ada A6000 GPUs.

**Discriminative Task Setup.** The classification task involves 11 classes with different zero-shot settings per backbone. For OPT-125M, we train on 6 classes and infer zero-shot on 5. For LLaMA-1B, given its superior generalization, we train on only 2 classes and evaluate on the remaining 9, testing its zero-shot adaptability. We match the generated response to the closest class label using SequenceMatcher, a fuzzy string matching algorithm that measures textual similarity. The vision encoder is CLIP. All experiments were conducted on computing nodes equipped with 2 NVIDIA A100 or 2 NVIDIA Ada A6000 GPUs.

**Hyper-parameter Setting.** The key hyperparameter setting is shown in Table I.

*Table 5.* Hyperparameter settings for generative and discriminative tasks.

| Hyperparameter | Generative | Discriminative |
|---|---|---|
| Learning Rate | $1 \times 10^{-4}$ | $1 \times 10^{-4}$ |
| Max Input Length | 1024 | 512 |
| Max Output Length | 128 | 32 |
| Text Neighbor Sampling | 11 | 11 |
| Image Neighbor Sampling | 5 | 11 |
| Batch Size (per device) | 2 | 2 |
| Gradient Accumulation Steps | 16 | 16 |
| Visual Tokens per Image | 4 | 4 |
| LoRA Rank | 64 | 64 |
| Prefix Tuning Virtual Tokens | 20 | 20 |
| Attention Diffusion Steps | 2 | 2 |
| Number of MM-QFormer Block | 1 | 1 |
| Attention Diffusion $\alpha$ | 0.1 | 0.1 |
| Number of Attention Heads | 8 | 8 |
| Training Epochs (OPT-125M) | 50 | 5 |
| Training Epochs (LLaMA-1B) | 3 | 3 |

# J. Modeling Graph as a Standalone Modality on Discriminative Setting

To further evaluate the role of graph as an independent modality, we conduct additional experiments on node classification using the OPT-125M backbone. Following the generative protocol described in Section 4.3, we feed GCN-derived node embeddings into the LLM as soft prompts, in a manner consistent with visual and textual tokens. This allows a fair comparison between graph-only modeling and our proposed fusion strategy.

*Table 6.* Performance of modeling graph as a standalone modality in the discriminative setting.

| Method | ROUGE-L | Accuracy | Recall | Precision |
|---|---|---|---|---|
| Subgraph's T&I | 0.8144 | 99.85 | 83.25 | 83.33 |
| +Graph Token | 0.6648 | 99.96 | 89.91 | 89.98 |
| +Node Token | 0.6892 | 99.53 | 88.78 | 89.98 |
| Ours (Hop-Diffused) | **0.8282** | **100.00** | **100.00** | **100.00** |

These results reinforce our finding that directly modeling the graph as a separate modality brings limited gains. This may be due to semantic mismatches between GNN-derived embeddings and pretrained vision-language features. Our Hop-Diffused approach significantly outperforms the others, providing further evidence for the advantage of structure-aware generation as stated in Proposition 4.1.

## K. Impact of the Graph Density

We analyze the impact of graph density under the generative setting by varying the number of (text, vision) neighbors from sparse (5,2) to dense (15,8). Table 7 shows that our model exhibits stable performance across varying densities and benefits from richer structural context, with consistent improvements in generation metrics.

*Table 7.* Impact of graph density on Graph4MM in the generative setting with OPT-125M backbone.

| Density | BLEU-4 | ROUGE-L | CIDEr |
|---|---|---|---|
| Sparse (5,2) | 0.0788 | 0.4049 | 0.7795 |
| Medium (11,5) | 0.0800 | 0.4076 | 0.7831 |
| Dense (15,8) | **0.0809** | **0.4086** | **0.8018** |

## L. Simulation Study on Attention Diffusion vs GAT under Small $K$

To examine over-smoothing at small diffusion depths, we conduct a simulation study using the Cora dataset. We measure the Dirichlet energy of node embeddings from both Hop-Diffused Attention and GAT over $K = 0$ to $4$. As shown in Table 8, Hop-Diffused Attention preserves energy better, especially at small $K$, justifying our default choice of $K = 2$.

*Table 8.* Dirichlet energy comparison across diffusion layers.

| Method | $K = 0$ | $K = 1$ | $K = 2$ | $K = 3$ | $K = 4$ |
|---|---|---|---|---|---|
| Hop-Diffused Attention | 3.2445 | 3.0105 | 2.9504 | 2.8360 | 2.7843 |
| GAT | 3.2445 | 0.9775 | 1.1345 | 0.6129 | 0.5448 |

## M. Performance Comparison under Different Random Seeds

We evaluate model stability under three random seeds in the generative setting. Table 9 compares Graph4MM with MMGL and reports mean ± standard deviation for each metric.

*Table 9.* Performance under different random seeds in the generative setting.

| Method | BLEU-4 | ROUGE-L | CIDEr |
|---|---|---|---|
| Graph4MM | **0.07991** ± 0.00066 | **0.40725** ± 0.00028 | **0.78904** ± 0.00305 |
| MMGL | 0.07762 ± 0.00059 | 0.40504 ± 0.00076 | 0.76907 ± 0.00286 |

## N. Performance Comparison with Link Prediction Setting

To further assess generalization, we introduce a new link prediction task on the Amazon-Sports dataset from MM-Graph Benchmark (Zhu et al., 2024b). We sample 10k positive and negative node pairs and use an 8:1:1 train/val/test split. The OPT-125M backbone and hyperparameters follow our node classification setup. Table 10 shows that Graph4MM significantly outperforms all baselines, with an average gain of 7.7% over the second-best method.

*Table 10.* Link prediction performance on the Amazon-Sports dataset with OPT-125M backbone.

| Method | ROUGE-L | Accuracy | Recall | Precision |
|---|---|---|---|---|
| (PLM) Node's Text | 0.1563 | 52.35 | 51.87 | 72.67 |
| (PLM) Node's Subgraph | 0.2871 | 56.81 | 56.26 | 74.56 |
| (MMGL) Node's Text & Image | 0.5603 | 93.92 | 93.86 | 95.51 |
| (MMGL) Subgraph's Text & Image | 0.7352 | 95.46 | 95.44 | 94.35 |
| (Graph4MM) Hop-Diffused MM-QFormer | **0.8904** | **99.98** | **99.97** | **99.86** |

## O. Validation Accuracy Curve of OPT-125M During Training Discriminative Task

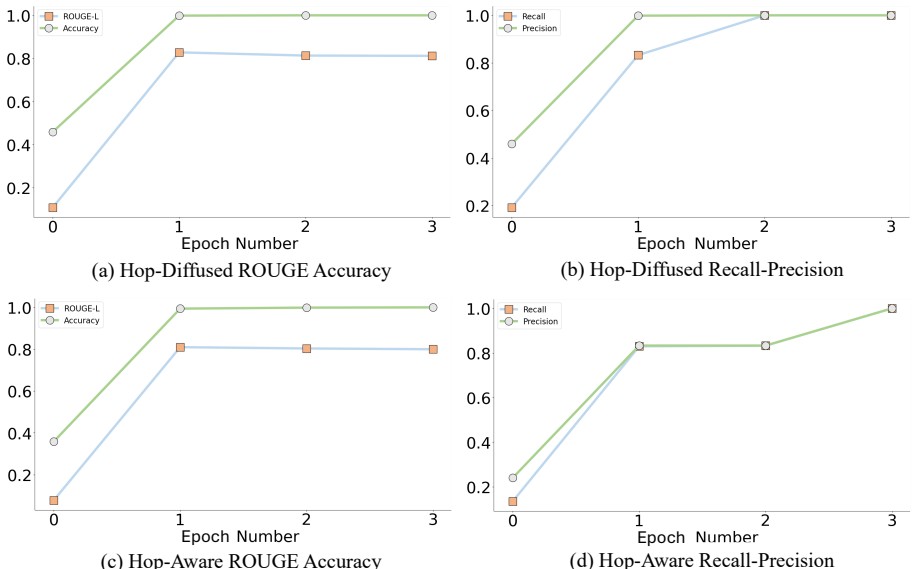

*Figure 4.* Validation performance across epochs in the discriminative task for Hop-Diffused and Hop-Aware methods on OPT-125M.

## P. Accuracy Evolution in Zero-Shot Node Classification with Different Number of Unseen Classes

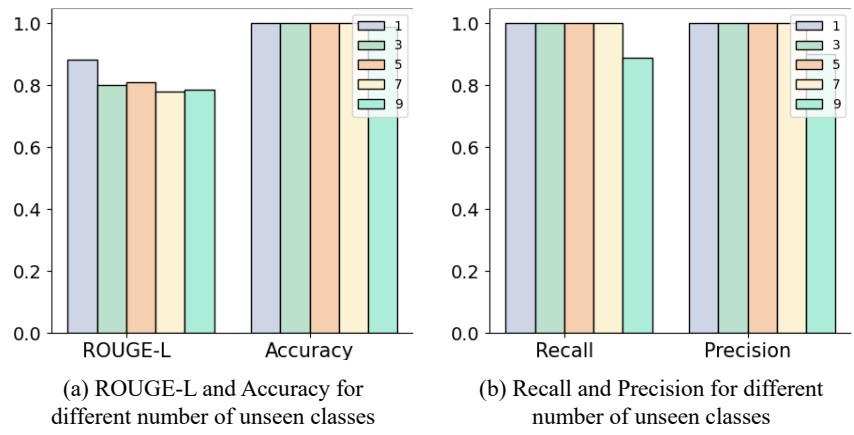

*Figure 5.* The performance in discriminative task with OPT-125M backbone across different number of unseen classes (1, 3, 5, 7, 9).

## Q. Example Prompt and Output

### Q.1. Example Prompt and Output in Generative Setting on WikiWeb2M Dataset

**Task:** Summarize the given section content.

Section Content: In 1682, he published a sermon titled "By what means can ministers best win souls?" and in 1692, a letter to a minister in the country—supposed to be his eldest brother, William (1640), minister of Borthwick, Midlothian—entitled "A Vindication of the Protestant Doctrine concerning Justification and of its Preachers and Professors from the unjust Charge of Antinomianism." This "angry letter," as Dr. Calamy calls it, was occasioned by the violent controversy that broke out among the dissenting ministers of London after the republication in 1690 of the works of Dr. Tobias Crisp. Donald Macleod called it "unrivalled." Charges of Antinomianism were made on one side and of Arminianism on the other, and Traill was distinguished for his zeal against Arminianism. A somewhat similar controversy, known as the Marrow Controversy, followed in Scotland, and as Boston of Ettrick and others took the same side as Traill, his works became very popular among them and their adherents. *He later published* "Sermons on the Throne of Grace from Heb. iv. 16" (3rd edit. 1731) and "Sermons on the Prayer of Our Saviour, John xvii. 24." These works were devout, plain, and edifying, and were highly favored by those who were attached to evangelical religion.
Context Information:
Section Image Caption: Rev Robert Traill's New Testament (1656)
1-Hop Neighbor Section Context: Robert Traill's early education was carefully superintended by his father, and at the University of Edinburgh he distinguished himself in both literary and theological studies. At the age of nineteen, he stood beside James Guthrie on the scaffold. He was later associated with John Welsh, minister of Irongray, known for holding armed conventicles. In 1666, he and his family were forced into hiding after a controversial book was found in their home. In 1667, he was denounced as a 'Pentland rebel' and fled to Holland, joining his exiled father and other Scottish refugees...
2-Hop Neighbor Section Context: Robert Traill was a church minister at Cranbrook in Kent. *He was born at Elie in Fife in 1642*. He was incarcerated on the Bass Rock, an island in the Firth of Forth, from July 19, 1677, to October 5, 1677. *His work* was often quoted by J. C. Ryle and is still published in the *21st century*.
(The remaining contexts are omitted for brevity.)

**Expected Output:**
His first short publication did not occur until he was *forty years old*, and the next did not appear until he was *fifty*.
**Graph4MM Output:**
The first work work was not appear until was thirty years old, it second publication not appear until he was fifty.

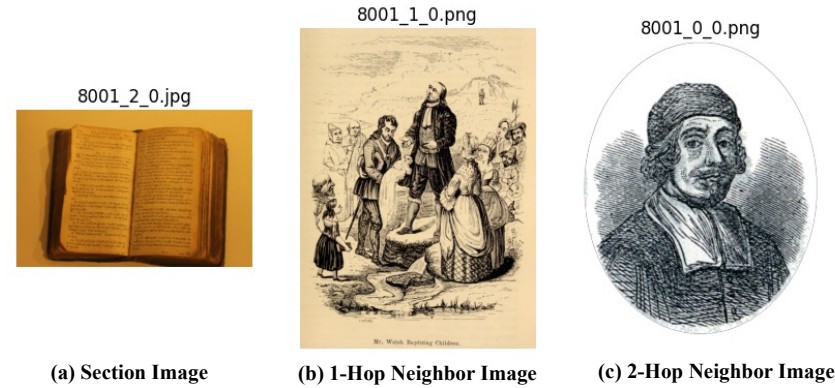

8001_2_0.jpg  8001_1_0.png  8001_0_0.png

(a) Section Image  (b) 1-Hop Neighbor Image  (c) 2-Hop Neighbor Image

*Figure 6.* Images from multi-hop neighboring sections within a webpage.

**Q.2. Example Prompt and Output in Discriminative Setting on Ele-Fashion Dataset**

**Task:** Classify the given item.

Node Description: Speedo Men's Sonic Warmup Jacket
Available Classes:
0: Sneakers and Men's Formal Shoes
1: Lingerie, Costumes, and Women's Footwear
2: Jewelry and Accessories
3: Stockings and Watches
4: Graphic T-Shirts and Sweatshirts
6: Scarves, Suspenders, and Wallets
7: Undergarments and Socks
8: Basic T-Shirts
9: Men's Casual and Formal Shirts
10: Dresses
11: Shoes and Boots
Neighbor Information:
**1-Hop Neighbor:** Speedo Women's Female Sonic Warm-Up Jacket
**2-Hop Neighbor:** Speedo Women's Female Sonic Warm-Up Pant
**2-Hop Neighbor:** Speedo Men's Sonic Warmup Jacket

**Expected Output:**
4:Graphic T-Shirts and Sweatshirts
**Graph4MM Output:**
4:Graphic T-Shirts and Sweatshirts.

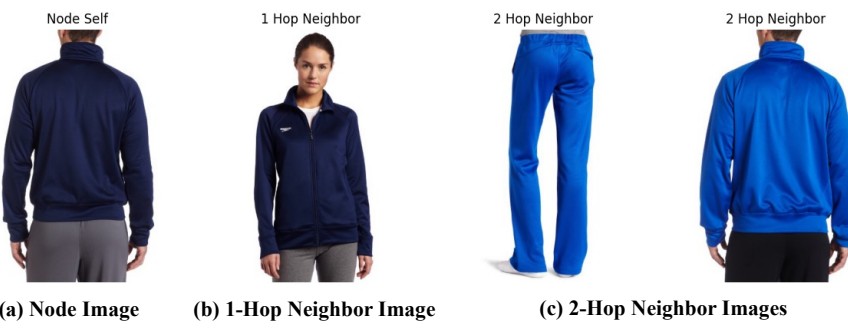

|                  |                        |                          |
|:----------------:|:----------------------:|:------------------------:|
| **(a) Node Image** | **(b) 1-Hop Neighbor Image** | **(c) 2-Hop Neighbor Images** |

*Figure 7.* Images from multi-hop neighboring nodes.

