# OpenReview forum: "Graph4MM: Weaving Multimodal Learning with Structural Information"
_ICML.cc/2025/Conference — ICML 2025 poster_

### Official Review · Reviewer_viW1 · 2025-03-13

**Overall Recommendation:** 3

**Summary:**

This paper introduces Graph4MM, a novel framework for multimodal learning that leverages graph structures to model complex relationships between text and images. The framework can be divided into two parts: Hop-Diffused Attention and MM-QFormer. By giving both theoretical and empirical analysis, the authors demonstrate the superiority of Graph4MM.

**Claims And Evidence:**

Most claims are well supported. But some points can be further improved.
1.	The paper claims that treating graph structure as a guide rather than a standalone modality is more effective, but the results in Table 4 only contain generative task, maybe discriminative task should also be included.
2.	The paper does not discuss computational efficiency trade-offs. Maybe results like runtime, memory usage comparison can be included.
3.	The paper uses WikiWeb2M and ELE-FASHION, but testing on more diverse domains would be better.
4.	There are no experiments about proposition 4.1, more empirical validation of this specific theory would be beneficial.

**Essential References Not Discussed:**

NO

**Experimental Designs Or Analyses:**

1.	The interaction between text and vision modalities should be taken into consideration in an ablation study.
2.	How does the random seed influence the results? Authors should claim this, make sure the results are not lucky initialization.
3.	The paper should systematically explore the influence of different graph densities or structures.

**Methods And Evaluation Criteria:**

Yes, it makes sense. But there are some points:
1.	The different setting for OPT-125M and LLaMA-1B(5 unseen classes vs 9 unseen classes)
2.	More diverse datasets would be better
3.	The concrete analysis for the computational efficiency should be included

**Other Comments Or Suggestions:**

NO

**Other Strengths And Weaknesses:**

NO

**Questions For Authors:**

NO

**Relation To Broader Scientific Literature:**

NO

**Theoretical Claims:**

The proofs are generally correct.

---

> ### Author Rebuttal · Authors · 2025-03-31
>
> Thanks very much for your valuable questions, each of them are quite actionable. We provide our response in the form of Q&A as follows.
>
> > **Claims And Evidence (CE) 1 & CE4**: Graph as a standalone modality in discriminative setting; empirical validation of Proposition 4.1.
>
> **A1**: We conducted additional experiments on node classification (OPT-125M backbone) to assess the effectiveness of modeling the graph as a standalone modality. Following the generative setting described in Lines 392–395, we use GCN-derived graph/node embeddings as soft prompts that serve as input to the downstream LLM, equivalent to vision and language tokens. The results are shown below.
>
> **Table 1**: Performance of "modeling graph as a standalone modality" in the discriminative setting.
> |Method|ROUGE-L|Accuracy|Recall|Precision|
> |------|-------|--------|------|----------|
> |Subgraph's T&I|0.8144|99.85|83.25|83.33|
> |+Graph Token|0.6648|99.96|89.91|89.98|
> |+Node Token|0.6892|99.53|88.78|89.98|
> |Ours (Hop-Diffused)|**0.8282**|**100.00**|**100.00**|**100.00**|
>
> These results reinforce our earlier finding: treating the graph as a separate modality does not lead to consistent gains, likely due to the semantic gap between GNNs with pretrained vision-language features. In contrast, our method performs significantly better, supporting Proposition 4.1 alongside Table 4 in our paper.
>
> ***
> > **CE2 & Methods And Evaluation Criteria (MEC) 3**: Computational Efficiency of our method.
>
> **A2**: Due to space limitations, we refer the reviewer to [Section "Weakness (W) 1" in our response to Reviewer xJRk](https://openreview.net/forum?id=FB2e8PV6qg&noteId=qC65trdQn8), where we theoretically and empirically show that our method shares the same order of time complexity as the baseline, with only mild runtime overhead.
>
> For the memory cost, in the generative setting with the OPT-125M backbone, our method adds 13M trainable parameters via Hop-Diffused MM-QFormer, which is less than 6% of MMGL's 229M total parameters. This overhead would become even smaller as the LLM backbone scales up, thus, the memory usage of Graph4MM remains manageable.
> ***
> > **CE3 & MEC2**: Testing on more diverse domains.
>
> **A3**: Due to space limitations, we refer the reviewer to [Section "W2" in our response to Reviewer xJRk](https://openreview.net/forum?id=FB2e8PV6qg&noteId=qC65trdQn8), where we validate the effectiveness and strong generalization ability of our method under a completely different domain.
> ***
> > **MEC1**: Different number of unseen classes or OPT-125M and LLaMA-1B backbone.
>
> **A4**: For the LLaMA-1B model, the performance gap among different baselines under the 5 unseen-class setting is relatively small, as all methods achieve high classification accuracy. Thus, we increased the task difficulty by expanding the number of unseen classes to 9 for comparison to highlight the different performance between methods. In Appendix K, we provide the performance variation under different numbers of unseen classes.
> ***
>
> > **Experiment Design or Analysis (ED) 1**: Ablation of vision and text interaction.
>
> **A5**: We ablate modality-specific structures and inputs by (1) removing structural heuristics from text or vision, and (2) removing the vision modality entirely. As shown in Table 2, removing structural heuristics for either vision or text modality degrades performance, the same as excluding visual information. This highlights the importance of both modalities and their corresponding structural guidance for effective multimodal fusion.
>
> **Table 2**: Modality interaction ablation results in the generative setting (OPT-125M).
> |Method Variant|BLEU-4|ROUGE-L|CIDEr|
> |-------------|------|--------|------|
> |Hop-Diffused MM-QFormer|**0.0800**|**0.4076**|**0.7831**|
> |-$t_{G'}$Hop-Diffused Attention|0.0786|0.4065|0.7765|
> |-$p_{G'}$Hop-Diffused Attention|0.0769|0.4044|0.7684|
> |-Vision Modality|0.0770|0.3992|0.7606|
> ***
> > **ED2**: Influence of the random seeds.
>
> **A6**: Under the generative setting with OPT-125M, we report the mean and standard deviation over 3 random seeds (0–2), and compare with the best MMGL baseline under the same setup.
>
> **Table 3**: Model performance across random seeds.
> |Method|BLEU-4|ROUGE-L|CIDER|
> |------|------|--------|------|
> |Graph4MM|**0.07991±0.00066**|**0.40725±0.00028**|**0.78904±0.00305**|
> |MMGL|0.07762±0.00059|0.40504±0.00076|0.76907±0.00286|
>
> ***
> > **ED3**: Discussion of the graph density.
>
> **A7**: We study graph density under the generative setting by varying (text, vision) neighbors from sparse (5,2), to medium (11,5), and dense (15,8). As shown in Table 5, our model remains stable across densities and benefits from more neighbors, consistently improving generation performance.
>
> **Table 4**: Impact of graph density on Graph4MM in generative setting (OPT-125M).
>
> |Density|BLEU-4|ROUGE-L|CIDEr|
> |-------|------|--------|------|
> |Sparse|0.0788|0.4049|0.7795|
> |Medium|0.0800|0.4076|0.7831|
> |Dense|0.0809|0.4086|0.8018|

---

### Official Review · Reviewer_gTEN · 2025-03-13

**Overall Recommendation:** 3

**Summary:**

The paper Graph4MM introduces a graph-based multimodal learning framework that integrates structural relationships into foundation models to improve multimodal understanding. Unlike previous methods that treat graphs as standalone modalities, Graph4MM incorporates Hop-Diffused Attention to model multi-hop connectivity and MM-QFormer for cross-modal fusion using learnable query tokens. The framework achieves an average improvement of 6.93% across generative and discriminative tasks, outperforming large VLMs, LLMs, and multimodal graph baselines. Theoretical and empirical analysis demonstrates that leveraging graph structure enhances multimodal learning beyond traditional one-to-one alignments.

**Claims And Evidence:**

Yes

**Essential References Not Discussed:**

Line 50 “To the best of our knowledge, MMGL (Yoon et al., 2023) is the state-of-the-art work that models modalities into graphs and obtains promising performance in the generation task, as compared to single-modal pre-trained language models and vision-language models.”

Should cite relevant papers working on the multimodal knowledge graphs and talking about the relationship between them and Graph4MM.

For example,  some most relevant papers mentioned in the survey:

Knowledge Graphs Meet Multi-Modal Learning: A Comprehensive Survey (submitted to arxiv on 8 Feb 2024)

or any other sota works working on this.

**Experimental Designs Or Analyses:**

Yes, I checked them all. However, this paper has some limitations in the selection of dataset and baseline. For the details, please refer to weaknesses.

**Methods And Evaluation Criteria:**

Yes

**Other Comments Or Suggestions:**

No

**Other Strengths And Weaknesses:**

Strengths

1. Structured Multimodal Integration: Effectively incorporates multi-hop graph connectivity into multimodal learning, improving intra- and inter-modal interactions.
2.  Advanced Attention Mechanism: Hop-Diffused Attention enhances multimodal reasoning by encoding structural dependencies while avoiding over-smoothing.
3. Good Empirical Results: Outperforms large VLMs, LLMs, and multimodal graph baselines, achieving a 6.93% average improvement across tasks.

Weaknesses

1. Lack of citations for relevant references.  Such as works related to multimodal knowledge graphs. I understand that this type of work is not a baseline for this paper, but it should be cited and briefly discussed in relation to this study.
2. The proposed module lacks sufficient innovation. The paper introduces MM-QFormer, but its architecture shows little novelty compared to traditional Q-Former. The main difference lies in incorporating structural prior knowledge into the input embeddings, a concept already explored in previous works such as the Translator module in GraphTranslator [1], significantly reducing the originality of the contribution.
3. Generalizability of the proposed method. The paper evaluates its performance on only one dataset for graph-related tasks, which raises concerns about the generalizability. Using a broader set of graph-based multimodal benchmarks [2] would strengthen the validity of the results and demonstrate the robustness of the approach across diverse datasets.
4. Incomplete baseline selection. The paper lacks a comprehensive baseline comparison, particularly with Graph Large Language Models related methods [3,4] that can handle text-attributed graphs. Replacing embeddings with unified representations incorporating both text and image information can also show reasoning capabilities in multimodal graph tasks. However, the paper does not sufficiently explore or evaluate these alternatives, limiting the depth of its baseline analysis.


[1] GraphTranslator: Aligning Graph Model to Large Language Model for Open-ended Tasks (https://arxiv.org/pdf/2402.07197)

[2] Multimodal Graph Benchmark. (https://arxiv.org/pdf/2406.16321)

[3] LlaGA: Large language and graph assistant (https://arxiv.org/abs/2402.08170)

[4] Can we Soft Prompt LLMs for Graph Learning Tasks? (https://arxiv.org/abs/2402.10359)

**Questions For Authors:**

1. How does it improve upon similar ideas, such as the Translator module in GraphTranslator? MM-QFormer seems to just move it over and use it.
2. Given that the method is only evaluated on a single dataset for graph-related tasks, how can the authors ensure its robustness and generalizability across different multimodal graph benchmarks?
3.  Why were Graph Large Language Models related methods (or GNN-based approaches) that integrate text and image embeddings not included as baselines? How would the proposed method compare to such models in multimodal graph reasoning tasks?
4. The baseline MMGL is evaluated under both frozen and fine-tuned settings, whereas Graph4MM is only tested in the fine-tuned setting.Would a frozen version of Graph4MM still demonstrate effectiveness compared to MMGL?

**Relation To Broader Scientific Literature:**

1. Beyond One-to-One Vision-Language Models: Unlike BLIP-2 and Flamingo, which align single image-text pairs, Graph4MM models many-to-many multimodal relationships using graph structures.
2. Advancing Multimodal Graph Learning: Extends MMGL by integrating graph topology into multimodal fusion rather than treating it as a standalone modality.
3. Improving Graph-Based Attention: Builds on GATs and diffusion-based methods by introducing Hop-Diffused Attention, which encodes multi-hop connectivity while mitigating over-smoothing.
4. Enhancing Query-Based Transformers: Extends QFormer (used in BLIP-2) by incorporating structural priors, improving cross-modal alignment with structural guidance.

**Theoretical Claims:**

Yes, I checked them all and found no obvious errors.

---

> ### Author Rebuttal · Authors · 2025-03-31
>
> Thanks very much for your constructive questions, each of them are quite actionable. We provide our response in the form of Q&A as follows.
>
> > **Weakness (W) 1**: Should cite relevant papers working on the multimodal knowledge graphs and talking about the relationship between them and Graph4MM.
>
> **A1**: We thank the reviewer for the suggestion. The survey [1] covers a wide range of MMKG research, including multimodal knowledge graph construction, multimodal representation learning, and cross-modal reasoning. These topics are relevant to our work in terms of constructing multimodal graphs and inter-modality fusion. We will update our Related Work section to include the mentioned survey [1] as well as representative recent works such as MoMoK [2] and MarT [3].
>
> [1] Chen, Zhuo, et al. "Knowledge graphs meet multi-modal learning: A comprehensive survey." arXiv preprint 2024.
>
> [2] Zhang, Yichi, et al. "Multiple Heads are Better than One: Mixture of Modality Knowledge Experts for Entity Representation Learning." ICLR 2025.
>
> [3] Zhang, Ningyu, et al. "Multimodal analogical reasoning over knowledge graphs." ICLR 2023.
>
> ***
> > **W2 & Question (Q) 1**: How does Graph4MM compared to Q-Former and GraphTranslator?
>
> **A2**:
> * **Compared to Q-Former**:
>
> We would like to clarify that we do not claim the Q-Former architecture itself as our core contribution. Q-Former, originally from BLIP-2, is a standard module for visual-language fusion. Our key contribution lies in how we integrate structural heuristics into the multimodal fusion process.
>
>  * **Compared to GraphTranslator**:
>
> Our method does not use the Translator module in GraphTranslator. The only similarity between our work and GraphTranslator lies in the use of the Q-Former architecture for cross-modality fusion, as discussed in the previous point. The Translator module in GraphTranslator is specifically designed for text-attributed graphs, fusing GNN-derived graph embeddings and explicit textual descriptions of graph structure using the Q-Former architecture.
>
> In contrast, our method: (1) Escalates to full multimodal learning, incorporating vision in addition to text and graph structure. (2) Takes a fundamentally different approach to structural integration: instead of relying on explicit textual descriptions to align graph embeddings with the LLM’s space (as in GraphTranslator), we introduce hop-diffused attention—a sparse mechanism that implicitly encodes graph topology in the hidden state during vision-text fusion. (3) Provides both theoretical analysis and empirical evidence demonstrating that treating graphs as a separate modality, as done in GraphTranslator, is sub-optimal in complex multimodal scenarios where vision and language co-occur.
>
> We believe our contributions are significant and fundamentally different from existing methods.
>
> ***
>
> > **W3 & Q2**: Generalization of our method on different datasets.
>
> **A3**: Due to space limitations, we refer the reviewer to [Section "W2" in our response to Reviewer xJRk](https://openreview.net/forum?id=FB2e8PV6qg&noteId=qC65trdQn8), where we validate the effectiveness and strong generalization ability of our method under a new link prediction setting.
>
> ***
> >**W4 & Q3**: Compared to other Graph Large Language Model-based methods.
>
> **A4**: We did not include graph-LLM methods as baselines because most prior works operate in a unimodal (text-only) setting and are not designed to handle multimodal inputs involving images. To better demonstrate the effectiveness of our method in multimodal scenarios, we conducted two additional experiments:
>
> (a) We used CLIP-ViT to extract multimodal node embeddings as input to enhance LLaGA;
>
> (b) We enhance GraphPrompter by incorporating visual tokens in the same way as MMGL.
>
> For a fair comparison, all models were evaluated under the same generative setting and applied the same fine-tuning strategy using the OPT-125M backbone. The results are shown below.
>
> **Table 1**: Comparison with multimodal extensions of Graph-LLM baselines under the generative setting (OPT-125M).
>
> | Method           | BLEU-4 | ROUGE-L | CIDEr  |
> |------|-----|-------|------|
> | LLaGA            | 0.0642 | 0.3738  | 0.6628 |
> | GraphPrompter    | 0.0782 | 0.4045  | 0.7651 |
> | Ours (Hop-Diffused) | **0.0800** | **0.4076** | **0.7831** |
>
> These results further validate the effectiveness of our proposed approach, which introduces hop-diffused structural heuristics into the latent space of multimodal fusion, leading to superior performance in multimodal graph reasoning.
>
> ***
> > **Q4**: MMGL is evaluated under both frozen and fine-tuned settings?
>
> **A5**: We would like to clarify that all MMGL variants we compared were fine-tuned, following the official MMGL implementation. We provide additional results under the frozen generative setting with OPT-125.
>
> **Table 2**: Comparison on frozen backbone.
> |Method|BLEU-4|ROUGE-L|CIDEr|
> |--|--|---|--|
> |MMGL|0.0606|0.3382|0.6332|
> |Graph4MM|**0.0733**|**0.4018**|**0.7428**|

---

> > ### Comment · Reviewer_gTEN · 2025-04-05
> >
> > Dear authors,
> >
> > Thank you for your dedicated work. Most of my concerns are addressed, so I'd like to increase my score to 3.

---

> > > ### Author Response · Authors · 2025-04-05
> > >
> > > Dear Reviewer gTEN,
> > >
> > > Thank you for your thoughtful comments and for raising the score!
> > > We’re glad to hear that our responses helped address your concerns. We believe that these discussions have helped us better clarify the paper, highlight our core contributions, and further improve the quality of Graph4MM. We will ensure that all of these points are carefully reflected in the final camera-ready version.
> > >
> > > We sincerely appreciate your time and constructive feedback.
> > >
> > > Best regards,
> > >
> > > #8515 Authors

---

### Official Review · Reviewer_xJRk · 2025-03-14

**Overall Recommendation:** 4

**Summary:**

The paper proposes Graph4MM, a graph based multimodal learning framework. It integrates multi hop structural information into foundation models and fuses modality specific information. The main algorithmic ideas include Hop Diffused Attention and MM QFormer. Experiments show that Graph4MM outperforms larger VLMs, LLMs, and multimodal graph baselines, achieving a 6.93% average improvement.

**Claims And Evidence:**

The claims are generally supported by evidence. The authors conduct experiments on two datasets with various baselines. The performance improvements in both generative and discriminative tasks demonstrate the effectiveness of Graph4MM.

**Essential References Not Discussed:**

There do not seem to be any essential references not discussed.

**Experimental Designs Or Analyses:**

The experimental designs are sound. The authors compare Graph4MM with multiple baselines under different input settings. They also conduct ablation studies to analyze the importance of structural information. However, the generalization of the results could be further verified by testing on more diverse datasets.

**Methods And Evaluation Criteria:**

The proposed methods make sense for the problem. Graph4MM uses graph based structures to model multimodal relationships, which is suitable for handling complex multimodal data. The evaluation criteria, including using relevant datasets like WIKIWEB2M and ELE FASHION for generative and discriminative tasks respectively, are appropriate for assessing the model's performance.

**Other Comments Or Suggestions:**

none

**Other Strengths And Weaknesses:**

A strength of the paper is its innovative approach to multimodal learning using graph structures, which shows significant performance improvements.

A weakness could be that the framework's complexity might limit its scalability. Also, the performance improvement might be dataset - specific, and more extensive testing is needed.

**Questions For Authors:**

none

**Relation To Broader Scientific Literature:**

The key contributions of the paper are related to the broader scientific literature. It builds on existing works in vision language models and multimodal graph neural networks.

**Theoretical Claims:**

The theoretical claims, such as the analysis of Hop Diffused Attention's properties and its comparison with GAT in terms of over smoothing, are presented with proofs.

---

> ### Author Rebuttal · Authors · 2025-03-31
>
> Thanks very much for your constructive questions, each of them are quite actionable. We take them quite seriously and prepare the following QA formatted response.
>
> >**Weakness (W) 1**: A weakness could be that the framework's complexity might limit its scalability.
>
> **A1**: We analyze the complexity of Graph4MM from both time complexity and runtime, showing that it shares the same order of complexity as LLM-based baselines like MMGL, with only a modest runtime overhead introduced by the Hop-Diffused MM-QFormer.
>
> Specifically, the complexity of our Hop-Diffused Attention is $O(|\mathcal{V}|^2 \cdot d + K \cdot |\mathcal{E}| \cdot d)$, and the Q-Former operates at $O(|\mathcal{V}|^2 \cdot n_q^2 \cdot d)$, where $|\mathcal{V}|$ is the number of nodes, $|\mathcal{E}|$ is the number of edges, $n_q$ is the number of multimodal queries, and $d$ is the hidden dimension. However, the LLM remains the dominant cost which scales up to $O(|\mathcal{V}|^2 \cdot T^2 \cdot d)$ due to processing per-node textual inputs—similar to MMGL, where $T$ is the average token length per node (usually $T \gg n_q$ ). We also provide per-batch training and inference time comparisons in Table 2, where the runtime shows only a mild increase.
>
> **Table 1**: Per-batch training and generation time comparison for generative setting on OPT-125M backbone (Mean ± Standard Deviation).
> | Method   | Secs per Batch (Training) | Secs per Batch (Generation) |
> |----------|----------------------------|------------------------------|
> | Graph4MM | 0.2185 ± 0.1259            | 0.0679 ± 0.0002              |
> | MMGL     | 0.1649 ± 0.1033            | 0.0470 ± 0.0002              |
>
> All LLM-based graph learning methods—including ours and MMGL—have quadratic complexity and are typically limited to small graphs, but they provide strong zero-shot generalization capabilities beyond the reach of traditional GNNs.
>
> ***
>
> >**W2**: Also, the performance improvement might be dataset-specific, and more extensive testing is needed.
>
> **A2**: To further validate the effectiveness and generalization of our method, we introduce a new discriminative task—link prediction—on the Amazon-Sports dataset from the Multimodal Graph Benchmark (MM-Graph Bench) [1]. We randomly sample 10k source-target node pairs with equal numbers of positive and negative samples, and split the dataset into train/validation/test sets in an 8:1:1 ratio. To ensure a fair comparison, all methods use the OPT-125M backbone. The hyperparameter settings follow those used in our discriminative node classification experiments. We compare our method against the most representative and promising baselines used in our main paper, including: (a) pretrained language model using nodes' and subgraphs' text, (b) MMGL’s best-performing methods with both text and image modalities, \(c) our Graph4MM with Hop-Diffused MM-Qformer.
>
> The results, shown in Table 2, demonstrate that Graph4MM significantly outperforms all baselines, achieving an average improvement of 7.7% over the second-best method across all evaluation metrics. This further demonstrates the strong generalization ability of our method, showing its applicability across different datasets and scenarios.
>
> **Table 2**: Performance comparison in Link Prediction setting on Amazon-Sports dataset (OPT-125M Backbone).
>
> | Method                                | R-L     | Acc (%) | Rec (%) | Pre (%) |
> |---------------------------------------|---------|---------|---------|---------|
> | (PLM) Node's Text                     | 0.1563  | 52.35   | 51.87   | 72.67   |
> | (PLM) Node's Subgraph                 | 0.2871  | 56.81   | 56.26   | 74.56   |
> | (MMGL) Node's Text & Image            | 0.5603  | 93.92   | 93.86   | 95.51   |
> | (MMGL) Subgraph's Text & Image        | 0.7352  | 95.46   | 95.44   | 94.35   |
> | (Graph4MM) Hop-Diffused MM-QFormer    | **0.8904**  | **99.98**  | **99.97**  | **99.86**  |
>
> [1] Zhu, Jing, et al. "Multimodal graph benchmark." arXiv preprint arXiv:2406.16321 (2024).

---

> > ### Comment · Reviewer_xJRk · 2025-04-08
> >
> > Thanks for the authors' response. After reading the rebuttal and the other reviewers' comments, I’m willing to raise my score.

---

> > > ### Author Response · Authors · 2025-04-08
> > >
> > > Dear Reviewer xJRk,
> > >
> > > Thank you very much for your follow-up and for raising the score! We sincerely appreciate your response and the time you have taken to review our rebuttal. We will ensure that all points from the rebuttal are carefully reflected in the final camera-ready version.
> > >
> > > Best regards,
> > >
> > > #8515 Authors

---

### Official Review · Reviewer_MweU · 2025-03-14

**Overall Recommendation:** 3

**Summary:**

The paper presents Graph4MM, for modelling multi-hop relationship within and between texts and images, modelled as an undirected graph. This approach enables the foundation model to be aware of the structural information (through the graph topology). They introduce Hop-Diffused MM-QFormer for incorporating multi-hop connectivity information and achieve SOTA results on generative (summarization) and discriminative (zero-shot fashion classification) task.

**Claims And Evidence:**

The authors provide a theoretical proof that GAT is more prone to over-smoothing effects but I doubt the proof (elaborated in Theoretical Claims). It would be nice to have it support it with a simulation study and spectral analysis.

**Essential References Not Discussed:**

NA

**Experimental Designs Or Analyses:**

The experiments are well designed, results carefully analyzed and experiment details provided in the supplementary

**Methods And Evaluation Criteria:**

Yes, the benchmarks make sense and the authors present a thorough comparison across models with ablations.

**Other Comments Or Suggestions:**

Some minor typos:

1. For proposition C.1, the authors use index $i$ for both -- row indexing and diffusion step indexing
2. Fig 2.: $t_{v_i}$ instead of $t_{i}$ and  $p_{v_i}$ instead of $v_{i}$ for consistency
3. L356R "shown in" in place of "shownin"

**Other Strengths And Weaknesses:**

Strengths:
1. The paper not only attempts at achieving SOTA for the tasks but also provides general result of using pretrained vision-language representations and also multi-hop representations
2. The designed MM-QFormer incorporates multimodal information efficiently, and can be applied to other use cases as well
2. The paper is easy to read and well-presented


Weaknesses:

Aside from my reservations about the theoretical claims, I have following doubts:
1. The authors should include a comparison with GAT to make sure that the prowess of their method comes from (1) their way of graph modelling alone, or (2) diffusion/hop-aware attention alone, or (3) both.
2. The computational complexity of the method might impede its scalability
3. Comparison on only one generative and one discriminative task might not be enough

I would be willing to increase my score if the authors are able to answer all my questions satisfactorily.

**Questions For Authors:**

1. Can the authors provide representative examples where GraphMM succeeds and MMGL fails?
2. Can you verify Table 2, row 5 (BLEU score for hop-aware -$t_{G'}$? Why is it significantly high?
3. Could you elaborate on R358-362? What do you mean by "modality-specific structural encoding"?

**Relation To Broader Scientific Literature:**

The paper is surely interesting because it not only achieves SOTA results on the two benchmarks but it also contains interesting findings ans results about incorporating the topological structure of the data for generation tasks

**Theoretical Claims:**

1. Rather than applying masking *before* softmax, the authors decide to apply it apply it after softmax. That is, $A_{i, j} = M_{i,j} \cdot A_{i,j}'$, where $A'$ is a row stochastic matrix (because of softmax function) and $M$ is a binary mask. This means that $A$ is not necessarily a row stochastic matrix. Why did the author go for such design choice rather than the conventional choice of masking before softmax (which was also done in Wang et al for attention diffusion? \
For proposition A.1 it relies on row stochasticity, but the $A$ is no longer row stochastic, or am I misinterpreting something?

2. I am not convinced with Proposition C.2 for GAT. Could you elaborate how: \
a. the first order Taylor expansion (about which point?) yields the said softmax approximation  \
b. is the softmax assumption valid? Because effectively, you approximate all the softmax functions to be identity (L747) which doesn't seem very valid.

3. While the authors perform theoretical analysis for infinite diffusion steps/GAT Layers, they keep the number of diffusion steps to be 2. For lower values of k, I am not sure how diffusion is better than GAT (assuming the correctness of Dirichlet Energy proofs)

---

> ### Author Rebuttal · Authors · 2025-04-01
>
> Thanks very much for your constructive questions. We provide our response in the form of Q&A.
>
> > **Theoretical Claim (TC) 1**: Softmax after masking.
>
> In our implementation, we do apply row-wise softmax after causal masking as $A_{i:}= \text{Softmax}(M_{i:} \odot A'_{i:})$. This ensures that $A$ is row-stochastic and compatible with our analysis in Proposition A.1. We will revise equations in line 212 for clarity, provide pseudocode in the Appendix, and release our code upon publication.
> ***
> > **TC2**: Elaborate the approximation in Proposition C.2 for GAT.
>
> a. To strengthen our proof, we adopt a more general assumption inspired by [1] that avoids the approximation with Taylor expansion. Specifically, we assume the activation function satisfies $0 \le \frac{\sigma(x)}{x} \le 1$ for $x \ne 0$ and $\sigma(0) = 0$, which holds for common activations such as ReLU and LeakyReLU. We define $\sigma(0)/0 := \sigma'(0)$ or 1 if $\sigma'(0)$ is undefined. Under this, any activation $\sigma(y)$ can be rewritten as $\sigma(y) = \mathrm{diag}\left( \frac{\sigma(y)}{y} \right) y$.
>
> b. We further generalize our proof as follows by removing the previous identity-matrix-based approximation and reach the same conclusion. We denote $D_i^{(k)}$ as the diagonal transformation matrix induced by the activation at layer $k$, satisfying $\text{diag}(0) \preceq D_i^{(k)} \preceq \text{diag}(1)$. The representation $X_i^{(k+1)}$ can be recursively expanded as: $ X_{.i}^{(k+1)} = \sum_{j_{k+1}=i, (j_k, ..., j_0) \in [d]^{k+1}} ( \prod^k_{l=0} W_{j_lj_{l+1}}^{(l)} ) D_{j_{k+1}}^{(k)} A^{(k)} \cdots D_{j_1}^{(0)} A^{(0)} X_{.j_0}^{(0)}$,  where $A^{(k)}$ is a row-stochastic aggregation matrix. Since all $D^{(k)}$ and $A^{(k)}$ have spectral norm less than or equal to 1, and each weight matrix $W^{(k)}$ is bounded, it follows that $\||X^{(k)}\|| \leq c^k \cdot \||X^{(0)}\||$ for some constant $0 < c < 1$. The Dirichlet energy is given by: $\mathcal{E}(X^{(k)}) = \frac{1}{n} \text{Tr}\left( X^{(k)\top} L X^{(k)} \right)$, where $L$ is the graph Laplacian. Applying the Rayleigh-Ritz Inequality yields: $\mathcal{E}(X^{(k)}) = O(\gamma^k \cdot \mathcal{E}(X^{(0)}))$ for some $\gamma \in (0, 1)$, indicating exponential decay of energy.
>
> [1] Demystifying Oversmoothing in Attention-Based Graph Neural Networks. NeurIPS 2023.
> ***
> > **TC3**: Attention Diffusion vs GAT under small $k$ for over-smoothing.
>
> To examine the over-smoothing behavior under small $k$, we further conducted a simulation study on the Cora dataset. We report the Dirichlet Energy of representations produced by GAT/Hop-diffused Attention across $k=0$ to $4$ in Table 1. (1) As $k$ increases, Hop-Diffused Attention shows slower decay than GAT. (2) Additionally, the Dirichlet Energy gap of two methods is already evident at $k=1$, suggesting that our choice of $k=2$ in the main experiments is valid.
>
> **Table 1**: Simulation results for Dirichlet Energy comparison.
> |Method|K=0|K=1|K=2|K=3|K=4|
> |---|---|----|----|----|----|
> |Hop-Diffused Attention|3.2445|3.0105|2.9504|2.8360|2.7843|
> |GAT|3.2445|1.2307|1.0293|0.8027|0.3730|
>
> ***
>
> >**Weakness (W) 1**: Disengage effects of graph modeling and Hop-diffused attention.
>
> Table 2 shows that (1) our graph modeling—adding multimodal neighbors—improves performance (Row 1→2), and (2) only hop-aware attention (Row 2→4) yields consistent gains over GAT-based structure modeling (Row 2→3), confirming the effectiveness of both components.
>
> **Table 2**: Ablation on graph and structural modeling (OPT-125M).
>
> |Method|BLEU-4|ROUGE-L|CIDEr|
> |------|---|----|----|
> |Node's T & I|0.0643|0.3825|0.6371|
> |Subgraph's T & I|0.0788|0.4051|0.7790|
> |GAT for Graph Tokens|0.0796|0.4052|0.7736|
> |Ours(Hop-Diffused)|**0.0800**|**0.4076**|**0.7831**|
>
>
> ***
> >**W2**: Computational complexity of Graph4MM.
>
> Due to space limits, please see [Section W1 in our response to Reviewer xJRk](https://openreview.net/forum?id=FB2e8PV6qg&noteId=qC65trdQn8). We show our method has the same complexity order as the baseline and mild runtime overhead.
> ***
>
> > **W3**: Should add new setting for comparison.
>
> Due to space limits, please see [Section W2 in our response to Reviewer xJRk](https://openreview.net/forum?id=FB2e8PV6qg&noteId=qC65trdQn8) for complete comparison results under a new setting.
>
> ***
> >**Q1**: Representative examples where GraphMM succeeds and MMGL fails.
>
> Below is a representative case where MMGL confuses node vs. neighbor features, while our structure-guided fusion yields the correct prediction.
> [Link to the example.](https://anonymous.4open.science/r/Graph4MM-3F30/representative%20example.png)
> ***
>
> >**Q2**: Why is BLUE in Table 2 high?
>
> BLEU-4 measures exact n-gram overlap. The inflation is due to the prediction containing more exact matches of domain-specific phrases from the ground truth.
> ***
> >**Q3**: Meaning of "modality-specific structural encoding".
>
> It refers to applying different structural modeling (Hop-Diffused / Hop-Aware) to each modality (vision or text).

---

> > ### Comment · Reviewer_MweU · 2025-04-04
> >
> > Dear authors, thanks for answering all my queries. I am satisfied with the answers.
> > I do agree with the fellow reviewers on similarity to previous works like QFormer and limited benchmarking, I am now positive about Graph4MM. Therefore, I increase my score to 3.
> >
> > Regards.

---

> > > ### Author Response · Authors · 2025-04-04
> > >
> > > Dear Reviewer MweU,
> > >
> > > Thanks very much for your reply!
> > >
> > > We are more than excited to learn of your satisfaction and appreciation for our rebuttal answers.
> > >
> > > Answering your raised questions, along with those from other reviewers, improves the paper's quality.
> > >
> > > We promise to add all rebuttal answers to the updated camera-ready version.
> > >
> > > Thanks again!
> > >
> > > #8515 Authors

---

### Decision · Program_Chairs · 2025-05-01

**Decision:**

Accept (poster)

**Comment:**

This paper received overall positive reviews. Reviewers found Graph4MM to be a novel and effective framework for integrating multi-hop structural information into multimodal learning via Hop-Diffused Attention and MM-QFormer. The approach shows strong empirical results on both generative and discriminative tasks. While some concerns were raised about theoretical proofs, baseline completeness, and evaluation scope, the authors provided a clear rebuttal that addressed these issues. The AC concurs with the reviewers and recommends acceptance. The authors are encouraged to expand evaluation and empirical support for theoretical claims in the camera-ready version.